# Exploiting Opponents under Utility Constraints in Sequential Games

**Martino Bernasconi-de-Luca**\*
Politecnico di Milano
martino.bernasconideluca@polimi.it

**Federico Cacciamani**\*
Politecnico di Milano
federico.cacciamani@polimi.it

**Simone Fioravanti**
Gran Sasso Science Institute
simone.fioravanti@gssi.it

**Nicola Gatti**
Politecnico di Milano
nicola.gatti@polimi.it

**Alberto Marchesi**
Politecnico di Milano
alberto.marchesi@polimi.it

**Francesco Trovò**
Politecnico di Milano
francesco1.trovo@polimi.it

## Abstract

Recently, game-playing agents based on AI techniques have demonstrated super-human performance in several sequential games, such as chess, Go, and poker. Surprisingly, the multi-agent learning techniques that allowed to reach these achievements do *not* take into account the actual behavior of the human player, potentially leading to an impressive gap in performances. In this paper, we address the problem of designing artificial agents that learn how to effectively exploit unknown human opponents while playing repeatedly against them in an *online* fashion. We study the case in which the agent's strategy during each repetition of the game is subject to constraints ensuring that the human's expected utility is within some lower and upper thresholds. Our framework encompasses several real-world problems, such as human *engagement* in repeated game playing and human education by means of *serious games*. As a first result, we formalize a set of linear inequalities encoding the conditions that the agent's strategy must satisfy at each iteration in order to do *not* violate the given bounds for the human's expected utility. Then, we use such formulation in an upper confidence bound algorithm, and we prove that the resulting procedure suffers from sublinear regret and guarantees that the constraints are satisfied with high probability at each iteration. Finally, we empirically evaluate the convergence of our algorithm on standard testbeds of sequential games.

## 1 Introduction

Algorithmic game theory and machine learning have recently contributed to groundbreaking achievements in artificial intelligence, leading to the deployment of artificial agents that defeated top human professionals in several recreational games. See, for example, the well-known milestones achieved in chess [Campbell *et al.*, 2002], Go [Silver *et al.*, 2016], and poker [Brown and Sandholm, 2018, 2019]. Surprisingly, multi-agent learning techniques that have been recently employed in such settings learn how to defeat humans without taking into account their actual behavior. Indeed, they learn strategies by simulating millions of plays in a self-play approach, without including any human player in the learning process. A direct effect of this methodology is that the resulting artificial agents do *not*

---

\*Equal contribution.

35th Conference on Neural Information Processing Systems (NeurIPS 2021).

adapt to the actual capabilities of humans, potentially leading to an impressive gap in performances compared to the case in which the agent learns strategies while taking them into account.

In this paper, we address the problem of designing artificial agents that learn how to effectively exploit unknown human opponents while playing repeatedly against them in an *online* fashion. In particular, we study the case in which the strategy that the agent plays during each repetition of the game is subject to constraints ensuring that the human's expected utility is within some lower and upper thresholds. Our framework models several real-world human-agent interactions, and it begets additional technical challenges compared to simple pure-exploitation scenarios.

One prominent application of our framework is when the artificial agent's goal is not only exploiting the human opponent, but also ensuring that he/she remains *engaged* in the game. Guaranteeing humans' engagement is crucial when designing artificial agents that play repeatedly against humans. Indeed, when playing against a super-human agent, most human players drop out from game playing, since they realize they are losing too often against it. As Egri-Nagy and Törmänen [2020] sharply observe, it is "*hopeless and frustrating to play against an AI, since it is practically impossible to win*". Different forms of engagement can be imagined (see, *e.g.*, [Abbasi *et al.*, 2017] for examples in computer games). In our framework, the human's engagement is modeled with the threshold constraints over his/her expected utility, with the following rationale. If the utility value falls under a given satisfaction threshold, then the human will get bored playing, as he/she loses too much and believes he/she has no hope to win. On the other hand, if such value raises above another given threshold, then the human will get bored since he/she is winning too often.

Another application scenario for our framework is that of *serious games* [Dörner *et al.*, 2016], whose purpose is to educate humans by asking them to perform tasks engagingly. If some tasks are excessively hard for the actual human's capabilities, then the human will give up the training of the entire set of tasks, since he/she is sure that he/she will never be able to address them. On the other hand, if some tasks are excessively easy, the human will give up the training since he/she is sure that he/she can solve all remaining tasks. Serious games are used in many different fields, such as, *e.g.*, military [DeFalco *et al.*, 2018], transportation [Rossetti *et al.*, 2013], urban planning [Poplin, 2011], and healthcare [Wang *et al.*, 2016], and can be modeled as general-sum games between a human learner and a computer teacher [Mayer, 2012].

**Original contributions.** We study two-player sequential (*i.e.*, extensive-form) games in which an artificial agent repeatedly plays against an unknown human opponent. We assume that the human has a fixed stochastic behavior and he/she does *not* learn over time. We do not make any structural assumption on the human's strategy, which requires the agent learning a probability distribution for each decision point of the human. While this is a crucial first step towards a more complex setting in which the human learns over time, the resulting model still presents several technical challenges that are worth to be investigated. First, we show how to derive, after each game repetition, a confidence region for the human's strategy such that his/her actual strategy stays within it with high probability. We show that such region is characterized by a set of linear constraints defined over the sequence-form strategy space. Notice that, during each game repetition, the agent only observes a partial sample of the human's strategy, made by the human's actions on the path in the game tree followed during play. By exploiting strong duality and the specific structure of the confidence region, we show that the thresholds constraints on the human's expected utility can be formulated as a set of linear inequalities, whose cardinality is linear in the size of the game tree. In particular, these constraints describe a subspace of the agent's sequence-form strategy space such that, for every possible human's strategy in the confidence region, the human's expected utility is within the given thresholds. We also derive a linear program with a linear number of constraints and variables to find the best agent's strategy satisfying the constraints. Then, we design an upper confidence bound algorithm, called COX-UCB, and we prove that it suffers from a sublinear regret and guarantees that the aforementioned constraints are satisfied with high probability at every iteration. Finally, we empirically evaluate the convergence of our algorithm on standard testbeds and show that our bounds are asymptotically tight. All the omitted proofs are provided in Appendices A, B, and C.

**Related works.** Our work is mainly related to opponent modeling, whose primary goal is to build models describing the behavior of one or multiple opponents from past interactions. Many methods are known in the literature. Specifically, Mealing and Shapiro [2017] and Foerster *et al.* [2018] propose a method to infer the parameters of the opponents' policies from the data collected during

past interactions. Instead, several other works propose methods that use beliefs over a fine set of opponents' types distinguishing for their behavior: Albrecht and Ramamoorthy [2014] treat types as black-box mappings, Albrecht and Stone [2017] infer parameters for the types, Barrett and Stone [2015] use deep learning to explicitly learn models, while He and Boyd-Graber [2016] do the same implicitly. These methods usually require huge amount of training data to be precise and adaptable to new opponents. In a recent work, Wu *et al.* [2021] propose a learning-to-exploit deep framework for implicit opponent modeling and an adversarial training procedure to automatically generate opponents so as to reduce the data needed for training. An approach that has more common ground with ours is the one adopted by Ganzfried and Sandholm [2011], who propose a game-theoretic approach to develop a deviation-based best-response algorithm. In particular, the work builds an opponent model based on the deviations between the opponent's strategy and a precomputed approximate equilibrium, and, then, computes a best response in real-time. The only works providing theoretical guarantees are [Ganzfried and Sandholm, 2015; Ganzfried and Sun, 2018], which deal with safe opponent exploitation, *i.e.*, guaranteeing a certain agent's payoff in expectation, given any opponent's strategy. Differently, in our setting, the goal is to guarantee a certain human's expected utility.

## 2 Preliminaries

In this section, we review the basic concepts and definitions related to sequential games that we need in the rest of this work (see the book by Shoham and Leyton-Brown [2008] for more details).

**Extensive-form games.** We focus on *two-player extensive-form games* (EFGs) with imperfect information in which an artificial agent faces a human opponent. We denote by $i$ the agent player, while $j$ is the human. Then, the set of players is $P \cup \{c\}$, where we let $P \coloneqq \{i, j\}$ and $c$ denotes a *chance player* that selects actions according to fixed known probabilities, representing exogenous stochasticity. An EFG is usually defined by means of a *game tree*, where $H$ is the set of nodes of the tree and $Z \subseteq H$ is the subset of terminal nodes, which are the leaves of the game tree. A node $h \in H$ is identified by the ordered list of actions encountered on the path from the root of the game tree to the node. Given a non-terminal node $h \in H \setminus Z$, we let $P(h) \in P \cup \{c\}$ be the unique player who acts at $h$ and $A(h)$ be the set of actions he/she has available. We let $u_i, u_j : Z \to \mathbb{R}$ be the payoff functions of players $i$ and $j$, respectively. Moreover, we denote by $p_c : H \to [0, 1]$ the function assigning each node $h \in H$ to the product of probabilities of chance moves on the path from the root of the game tree to $h$. Imperfect information is encoded by using *information sets* (infosets for short). A player $i$'s infoset $I$ groups nodes belonging to player $i$ that are indistinguishable for him/her, that is, $I \subseteq H \setminus Z$ is such that $P(h) = P(k) = i$ and $A(h) = A(k)$ for any pair of nodes $h, k \in I$. We let $\mathcal{I}$ be the set of player $i$'s infosets, which define a partition of the set of player $i$'s non-terminal nodes $\{h \in H \setminus Z \mid P(h) = i\}$. Moreover, with a slight abuse of notation, we let $A(I)$ be the set of actions available at all the nodes in infoset $I \in \mathcal{I}$. Analogously, we define $\mathcal{J}$ as the set of player $j$'s infosets, while, for any infoset $J \in \mathcal{J}$, we let $A(J)$ be the set of actions available at nodes in $J$. We focus on games with *perfect recall* in which infosets are such that no player forgets information once acquired.

**The sequence form of EFGs.** The *sequence form* is a compact way of representing EFGs with perfect recall [Von Stengel, 1996; Koller *et al.*, 1996], where the pure strategies of a player—specifying an action at each infoset of that player—are replaced by the concept of sequence. Any node $h \in H$ defines a *sequence* $\sigma_i(h)$ of player $i$, which is identified by the ordered list of player $i$'s actions on the path from the root of the game tree to $h$. In perfect-recall EFGs, all the nodes belonging to an infoset $I \in \mathcal{I}$ of player $i$ define the same player $i$'s sequence, which, by overloading notation, we denote by $\sigma_i(I)$. Sequence $\sigma_i(I)$ can be extended by appending any action $a \in A(I)$ available at $I$ at its end, obtaining another valid player $i$'s sequence that we denote as $\sigma_i(I)a$. Then, the set of player $i$'s sequences is $\Sigma_i \coloneqq \{\sigma_i(I)a \mid I \in \mathcal{I}, a \in A(I)\} \cup \{\varnothing\}$, where $\varnothing$ is the empty sequence (defined by all the nodes such that player $i$ never plays before them in the game tree). Analogously, we define $\Sigma_j \coloneqq \{\sigma_j(J)a \mid J \in \mathcal{J}, a \in A(J)\} \cup \{\varnothing\}$ as the set of all player $j$'s sequences. Mixed strategies in the sequence form are specified by defining the realization probability of each sequence. A *sequence-form strategy* of player $i$ is denoted by a vector $\boldsymbol{x} \in [0, 1]^{|\Sigma_i|}$, with $\boldsymbol{x}[\sigma_i]$ being the realization probability of sequence $\sigma_i \in \Sigma_i$.[2] To be well defined, a sequence-form strategy must

---

[2]In this work, we denote vectors by bold symbols. Given a finite set $S$ of dimension $|S| = d$, we denote by $\boldsymbol{v} \in \mathbb{R}^{|S|}$ a $d$-dimensional vector indexed over $S$, with $\boldsymbol{v}[s]$ being its component corresponding to $s \in S$.

satisfy a set of linear constraints, ensuring that realization probabilities of sequences encode a valid probability distribution over actions at each infoset. Formally, any $\boldsymbol{x} \in [0,1]^{|\Sigma_i|}$ must satisfy:

$$\boldsymbol{x}[\varnothing] = 1, \quad \text{and} \quad \boldsymbol{x}[\sigma_i(I)] = \sum_{a \in A(I)} \boldsymbol{x}[\sigma_i(I)a] \quad \forall I \in \mathcal{I}. \tag{1}$$

Constraints (1) can be written as $\boldsymbol{F}_i \boldsymbol{x} = \boldsymbol{f}_i$, where $\boldsymbol{F}_i \in \{-1, 0, 1\}^{(|\mathcal{I}|+1) \times |\Sigma_i|}$ and $\boldsymbol{f}_i \in \{0, 1\}^{|\mathcal{I}|+1}$ are a suitably-defined matrix and vector, respectively (see [Von Stengel, 1996] for their definitions). Analogously, we denote sequence-form strategies of player $j$ as vectors $\boldsymbol{y} \in [0,1]^{|\Sigma_j|}$ that satisfy the condition $\boldsymbol{F}_j \boldsymbol{y} = \boldsymbol{f}_j$, which is defined by linear constraints analogous to Constraints (1). For the ease of presentation, we let $\mathcal{X} := \{\boldsymbol{x} \in [0,1]^{|\Sigma_i|} \mid \boldsymbol{F}_i \boldsymbol{x} = \boldsymbol{f}_i\}$ and $\mathcal{Y} := \{\boldsymbol{y} \in [0,1]^{|\Sigma_j|} \mid \boldsymbol{F}_j \boldsymbol{y} = \boldsymbol{f}_j\}$ be the sets of all sequence-form strategies of players $i$ and $j$, respectively. Finally, given two strategies $\boldsymbol{x} \in \mathcal{X}$ and $\boldsymbol{y} \in \mathcal{Y}$, it is easy to check that player $i$'s expected payoff can be written as the bilinear form $\boldsymbol{x}^\top \boldsymbol{U}_i \boldsymbol{y}$, where $\boldsymbol{U}_i \in \mathbb{R}^{|\Sigma_i| \times |\Sigma_j|}$ is player $i$'s sequence-form *utility matrix*, defined as:

$$\boldsymbol{U}_i[\sigma_i, \sigma_j] := \sum_{z \in Z: \sigma_i(z) = \sigma_i \wedge \sigma_j(z) = \sigma_j} p_c(z) u_i(z) \quad \forall \sigma_i \in \Sigma_i, \forall \sigma_j \in \Sigma_j.$$

Analogously, the sequence-form utility matrix of player $j$ is $\boldsymbol{U}_j \in \mathbb{R}^{|\Sigma_i| \times |\Sigma_j|}$, and, thus, $\boldsymbol{x}^\top \boldsymbol{U}_j \boldsymbol{y}$ is his/her expected payoff given two sequence-form strategies $\boldsymbol{x} \in \mathcal{X}$ and $\boldsymbol{y} \in \mathcal{Y}$.

**Additional notation.** Given two sequences $\sigma_i, \sigma_i' \in \Sigma_i$ of player $i$, we write $\sigma_i \sqsubseteq \sigma_i'$ to denote that $\sigma_i$ is a *sub-sequence* of $\sigma_i'$; formally, this is the case whenever the ordered list of actions identified by $\sigma_i$ is a prefix of that of $\sigma_i'$. Similarly, we use notation $\sigma_j \sqsubseteq \sigma_j'$ for two sequences $\sigma_j, \sigma_j' \in \Sigma_j$ of player $j$. Moreover, given a player $i$'s sequence-form strategy $\boldsymbol{x} \in \mathcal{X}$ and a player $j$'s infoset $J \in \mathcal{J}$, we let $\rho_{-j}(J, \boldsymbol{x})$ be the probability of reaching $J$ given that player $j$ plays so as to reach it and player $i$ plays $\boldsymbol{x}$ (also accounting for chance probabilities); formally $\rho_{-j}(J, \boldsymbol{x}) := \sum_{h \in J} \boldsymbol{x}[\sigma_i(h)] p_c(h)$.

## 3 Exploiting opponents under utility constraints

We study settings in which the agent player $i$ repeatedly faces the human opponent $j$ in a two-player EFG. Our goal is the design of agents that learn the strategy of the human so as to effectively exploit it, while at the same time guaranteeing that the human's expected utility remains under control during the entire repeated interaction.[3] In the rest of this section, we formally introduce our problem and provide a general overview of the approach we undertake to tackle it.

We let $T$ be the number of times the EFG is played. Player $j$ plays according to the same sequence-form strategy $\boldsymbol{y}^* \in \mathcal{Y}$ at each iteration $t \in [T]$.[4] This strategy is unknown to player $i$. On the other hand, at iteration $t$, player $i$ selects and plays a strategy $\boldsymbol{x}^t \in \mathcal{X}$. Then, at the end of the iteration, player $i$ receives as feedback a sequence of player $j$'s actions $\sigma_j^t \in \Sigma_j$, which is defined by the path in the game tree followed during game playing at that iteration (notice that $\sigma_j^t$ is made of actions sampled according to player $j$'s strategy $\boldsymbol{y}^*$). In the following, for $t \in [T]$, we let $\mathcal{H}^t$ be the history of feedbacks received by the agent player $i$ up to iteration $t$ (included), namely $\mathcal{H}^t := (\sigma_j^1, \sigma_j^2, \dots \sigma_j^t)$.

To ensure that player $j$'s utility is kept under control during the repeated interaction, player $i$ must play strategies $\boldsymbol{x}^t$ such that the resulting player $j$'s expected utilities are within some given thresholds. The lower limit of the range ensures that the agent does *not* over-exploit the human. On the other hand, the upper limit of the range guarantees that the expected payoff of the human is *not* too high. Formally, we require $\boldsymbol{x}^t \in \mathcal{X}^t$, where the subset $\mathcal{X}^t \subseteq \mathcal{X}$ is defined as follows:

**Definition 1** (Utility-constrained Strategy Set). *Let $t \in [T]$ and $\delta \in (0, 1)$. Given a lower limit $\alpha \in \mathbb{R}$ and an upper limit $\beta \in \mathbb{R}$, we define the* utility-constrained strategy set $\mathcal{X}^t \subseteq \mathcal{X}$ *at iteration $t$ as the set of player $i$'s sequence-form strategies $\boldsymbol{x} \in \mathcal{X}$ such that $\mathbb{P}(\alpha \leq \boldsymbol{x}^\top \boldsymbol{U}_j \boldsymbol{y}^* \leq \beta) \geq 1 - \delta$, with respect to the randomness of the history $\mathcal{H}^{t-1}$ of feedbacks observed by player $i$ up to iteration $t - 1$ (included).[5]*

---

[3]The methodology that we propose in this paper can also be adapted to control the agent's expected utility, rather than the one of the human player.

[4]In this work, given $n \in \mathbb{N}_+$ we denote by $[n]$ the set $\{1, \dots, n\}$ of the first $n$ natural numbers.

[5]In the rest of this work, we make implicit the dependency of $\mathcal{X}^t$ from $\delta$, $\alpha$, and $\beta$, as the values of these parameters will be clear from context.

After $T$ game repetitions, given the strategies $\boldsymbol{x}^t \in \mathcal{X}^t$ played by the agent player $i$ during iterations $t \in [T]$, we measure his/her performance by means of the following notion of regret:

$$R^T := \sum_{t=1}^T \left[ \max_{\boldsymbol{x}^* \in \mathcal{X}^t} (\boldsymbol{x}^*)^\top \boldsymbol{U}_i \boldsymbol{y}^* - (\boldsymbol{x}^t)^\top \boldsymbol{U}_i \boldsymbol{y}^* \right],$$

which represents how much player $i$ would have gained in expectation by playing a utility-maximizing strategy in the utility-constrained strategy set $\mathcal{X}^t$ rather than $\boldsymbol{x}^t$, at each iteration $t \in [T]$. The goal that we pursue in the rest of this work is to achieve sublinear regret, that is $R^T = o(T)$, while at the same time guaranteeing that the played strategies $\boldsymbol{x}^t$ satisfy the constraints defined by the sets $\mathcal{X}^t$.

**Overview of our results.** In what follows, we give a brief sketch of the approach we adopt to tackle the problem. We propose a learning algorithm for the agent player $i$, which we call *Constrained Opponent eXploitation with Upper Confidence Bounds* (COX-UCB). It builds on two core components. The first one deals with the construction of the utility-constrained strategy set $\mathcal{X}^t$ at each iteration $t \in [T]$. It works by building a confidence region $\mathcal{Y}^{t-1} \subseteq \mathcal{Y}$ for player $j$'s strategy, using the history $\mathcal{H}^{t-1}$ of feedbacks observed up the previous iteration $t - 1$. This is such that the true (unknown) strategy $\boldsymbol{y}^*$ lies within $\mathcal{Y}^{t-1}$ with probability at least $1 - \delta$, for some fixed confidence level $\delta \in (0, 1)$. Then, the utility-constrained strategy set $\mathcal{X}^t$ can be characterized by a set of linear inequalities that exploits the structure of the confidence region $\mathcal{Y}^{t-1}$. A detailed formal treatment of this first component is provided in Section 4. The second core component consists in a rule to select the strategy $\boldsymbol{x}^t \in \mathcal{X}^t$ to play at each iteration $t \in [T]$. We propose an approach based on the *optimism in face of uncertainty* principle. More details on this second component can be found in Section 5, together with the regret bounds attained by the algorithm. We refer the reader to Algorithm 1 for a general sketch of our COX-UCB algorithm, where two alternative implementations of the procedure SELECTSTRATEGY($\mathcal{X}^t, \mathcal{Y}^{t-1}$) will be given in Algorithms 2 and 3 in Section 5.

---

**Algorithm 1** COX-UCB

---

1: $t \leftarrow 1$
2: **while** $t \leq T$ **do**
3:      Build confidence region $\mathcal{Y}^{t-1}$ from history of past feedbacks $\mathcal{H}^{t-1}$             $\triangleright$ Subsection 4.1
4:      Use $\mathcal{Y}^{t-1}$ to build the utility-constrained strategy set $\mathcal{X}^t$             $\triangleright$ Subsection 4.2
5:      $\boldsymbol{x}^t \leftarrow$ SELECTSTRATEGY($\mathcal{X}^t, \mathcal{Y}^{t-1}$)             $\triangleright$ Section 5
6:      Play the game according to strategy $\boldsymbol{x}^t$
7:      Observe player $j$'s sequence $\sigma_j^t$, obtained from the path in the game tree followed during play
8:      $\mathcal{H}^t \leftarrow \mathcal{H}^{t-1} \cup \{\sigma_j^t\}$
9:      $t \leftarrow t + 1$

---

## 4 How to control the human's expected utility

In this section, we provide the formal details on the construction of the utility-constrained strategy set $\mathcal{X}^t$ at each iteration $t \in [T]$ (Definition 1). We split the section into two main parts:

- Subsection 4.1 shows how to use the history $\mathcal{H}^t$ of feedbacks observed by player $i$ up to iteration $t$ to derive a confidence region $\mathcal{Y}^t \subseteq \mathcal{Y}$ for player $j$'s strategy $\boldsymbol{y}^*$, such that $\boldsymbol{y}^*$ lies within $\mathcal{Y}^t$ with probability at least $1 - \delta$ for some fixed confidence $\delta \in (0, 1)$;

- Subsection 4.2 describes how to exploit the confidence region $\mathcal{Y}^{t-1}$ built using feedbacks observed up to iteration $t - 1$ to construct a set of linear constraints that fully characterize sequence-form strategies in the utility-constrained strategy set $\mathcal{X}^t$ at iteration $t$.

All the proofs omitted from this section are reported in Appendices A and B.

### 4.1 Building a confidence region for the human's strategy

Let us recall that, at each iteration $t \in [T]$, player $i$ observes a player $j$'s sequence $\sigma_j^t$ determined by selecting actions to play during the game according to the sequence-form strategy $\boldsymbol{y}^*$. We build the desired high-probability confidence region $\mathcal{Y}^t$ by exploiting information provided by observed

sequences to derive, for each player $j$'s infoset $J \in \mathcal{J}$, appropriate confidence intervals for the realization probabilities $\boldsymbol{y}^*[\sigma_j(J)a]$ of sequences $\sigma_j(J)a$ terminating with an action $a \in A(J)$ at $J$.[6]

**The case of a single infoset.** Before showing our general technique, it is useful to present the easier setting in which player $j$ has a unique infoset, and, thus, his/her strategy is defined as a probability distribution $\boldsymbol{p} \in \Delta^{|A|}$, where, with an abuse of notation, $A$ denotes the finite set of actions available at the infoset. In this case, player $i$ observes $t$ actions $a^1, \ldots, a^t \in A$ sampled independently according to $\boldsymbol{p}$. Then, a natural estimator for $\boldsymbol{p}$ is the empirical frequency of actions $\boldsymbol{p}^t \in \Delta^{|A|}$, defined so that $\boldsymbol{p}^t[a] := \frac{1}{t} \sum_{\tau=1}^{t} \mathbb{1}\{a^\tau = a\}$ for every $a \in A$. By noticing that $t\,\boldsymbol{p}^t$ is a random variable following a multinomial distribution with parameters $t$ and $\boldsymbol{p}$, that is $t\,\boldsymbol{p}^t \sim \mathcal{M}(t; \boldsymbol{p})$, the following lemma by Devroye [1983] can be used to derive the desired confidence intervals for the probabilities $\boldsymbol{p}[a]$.[7]

**Lemma 1** (Lemma 3 by Devroye [1983]). *Let $\boldsymbol{p} \in \Delta^{|A|}$ and $a^1, \ldots, a^t \in A$ be $t$ actions sampled independently according to $\boldsymbol{p}$. Then, for any $0 < \delta \le 3 \exp(-4|A|/5)$, it holds:*

$$\mathbb{P}\left(\sum_{a \in A} \left|\boldsymbol{p}^t[a] - \boldsymbol{p}[a]\right| \le 5\sqrt{\frac{\ln(3/\delta)}{t}}\right) \ge 1 - \delta.$$

By exploiting the fact that $\boldsymbol{p} \in \Delta^{|A|}$, we can refine the result in Lemma 1 by giving bounds that hold for each component of $\boldsymbol{p}$ separately (see Lemma 2 below). This additional step is crucial when building confidence intervals for realization probabilities $\boldsymbol{y}^*[\sigma_j(J)a]$ at infoset $J$ in general.

**Lemma 2.** *Let $\boldsymbol{p} \in \Delta^{|A|}$ and $a^1, \ldots, a^t \in A$ be $t$ actions sampled independently according to $\boldsymbol{p}$. Then, for any $0 < \delta \le 3 \exp(-4|A|/5)$, it holds:*

$$\mathbb{P}\left(\bigcap_{a \in A}\left\{\left|\boldsymbol{p}^t[a] - \boldsymbol{p}[a]\right| \le \frac{5}{2}\sqrt{\frac{\ln(3/\delta)}{t}}\right\}\right) \ge 1 - \delta.$$

Next, we generalize the approach described above to the general case of any infosets structure.

First, we introduce some useful random variables. For every iteration $t \in [T]$, player $j$'s infoset $J \in \mathcal{J}$, and action $a \in A(J)$, we let $O^t(J, a) := \mathbb{1}\{\sigma_j(J)a \sqsubseteq \sigma_j^t\}$ be a random variable that is equal to 1 if and only if player $j$ played action $a$ at infoset $J$ during iteration $t$, while it is equal to 0 otherwise. It is easy to check that $O^t(J, a)$ follows a Bernoulli distribution with parameter $p^t(J, a) := \boldsymbol{y}^*[\sigma_j(J)a]\, \rho_{-j}^t(J)$, where, for the ease of presentation, we let $\rho_{-j}^t(J) := \rho_{-j}(J, \boldsymbol{x}^t)$ be the contribution to the probability of reaching infoset $J$ due to player $i$'s strategy $\boldsymbol{x}^t$ and chance probabilities.[8] The random variables $O^t(J, a)$ are instrumental for defining $N^t(J, a) := \sum_{\tau=1}^{t} O^\tau(J, a)$, which represents the number of times $a$ is played at $J$ up to iteration $t$. Intuitively, variables $N^t(J, a)$ of infoset $J \in \mathcal{J}$ play the same role as the random vector $t\,\boldsymbol{p}^t$ in the single-infoset case.

We follow an approach analogous to that of the single-infoset case at each player $j$'s infoset, and, then, put all the resulting confidence intervals together to define $\mathcal{Y}^t$. To do so, we need to circumvent the following issue: at each infoset $J \in \mathcal{J}$, the random variables $N^t(J, a)$ for $a \in A(J)$ are *not* jointly distributed as a multinomial, preventing a direct application of Lemma 1. We deal with this by adding a fictitious action at each infoset, so that random variables $N^t(J, a)$ are multinomially distributed for each $J \in \mathcal{J}$. Then, the fictitious action can be easily factored out by using Lemma 2.

Let $A_\diamond(J) := A(J) \cup \{a_\diamond\}$ be the new action set at $J \in \mathcal{J}$, with $a_\diamond$ denoting the fictitious action. For $t \in [T]$, we define $O^t(J, a_\diamond) := \mathbb{1}\{\sigma_j(J)a \not\sqsubseteq \sigma_j^t\}$ as a random variable equal to 1 if and only if

---

[6]Let us remark that deriving $\mathcal{Y}^t$ is made considerably challenging by the fact that, at each $t \in [T]$, only the sequence $\sigma_j^t$ of actions actually played by player $j$ is observed. On the other hand, if player $i$ would be able to observe the actual pure strategy selected by player $j$ at $t$, the problem would admit a much easier solution consisting in building a single confidence interval for player $j$'s average strategy.

[7]In this work, we denote by $\Delta^{|S|}$ the $(|S| - 1)$-dimensional simplex indexed over the finite set $S$. Moreover, we denote by $\mathbb{1}\{\cdot\}$ the indicator function for the event enclosed in curly braces, while $\mathcal{M}(n; \boldsymbol{v})$ denotes a multinomial probability distribution, where $n \in \mathbb{N}_+$ is the number of trials and $\boldsymbol{v} \in \Delta^{|S|}$ is a vector defining the probabilities of observing each element in the finite set $S$.

[8]In the rest of this work, we assume w.l.o.g. that $\rho_{-j}^t(J) > 0$ for any $t$ and $J$. This is possible thanks to the fact that strategies $\boldsymbol{x}^t$ selected by the algorithm proposed in Section 5 always ensure that such conditions hold.

it is *not* the case that $a$ is played at $J$ during iteration $t$ (this also includes all the cases in which $J$ is not reached), and 0 otherwise. Clearly, $O^t(J, a_\diamond)$ follows a Bernoulli distribution with parameter $p^t(J, a_\diamond) := 1 - \sum_{a \in A(J)} p^t(J, a)$. Moreover, we let $N^t(J, a_\diamond) := \sum_{\tau=1}^{t} O^\tau(J, a_\diamond)$. Then, for each player $j$'s infoset $J \in \mathcal{J}$, we define $\boldsymbol{n}_J^t \in \mathbb{N}^{|A_\diamond(J)|}$ as a random vector such that $\boldsymbol{n}_J^t[a] := N^t(J, a)$ for every $a \in A_\diamond(J)$. By letting $\bar{\rho}_{-j}^t(J) := \frac{1}{t} \sum_{\tau=1}^{t} \rho_{-j}^\tau(J)$, we have:

$$\mathbb{E}\left[N^t(J, a)\right] = \sum_{\tau=1}^{t} \mathbb{E}\left[O^\tau(J, a)\right] = \boldsymbol{y}^*[\sigma_j(J)a] \sum_{\tau=1}^{t} \rho_{-j}^\tau(J) = t \, \bar{\rho}_{-j}^t(J) \boldsymbol{y}^*[\sigma_j(J)a],$$

while it is easy to check that $\mathbb{E}\left[N^t(J, a_\diamond)\right] = t - t \, \bar{\rho}_{-j}^t(J) \sum_{a \in A(J)} \boldsymbol{y}^*[\sigma_j(J)a]$. As a result, we conclude that $\boldsymbol{n}_J^t$ follows a multinomial distribution, circumventing our initial issue. Formally:

$$\boldsymbol{n}_J^t \sim \mathcal{M}(t, \boldsymbol{v}), \boldsymbol{v} \in \Delta^{|A_\diamond(J)|} \text{ s.t. } \boldsymbol{v}[a] = \begin{cases} \bar{\rho}_{-j}^t(J) \boldsymbol{y}^*[\sigma_j(J)a] & \text{if } a \in A(J) \\ 1 - \bar{\rho}_{-j}^t(J) \sum_{a \in A(J)} \boldsymbol{y}^*[\sigma_j(J)a] & \text{if } a = a_\diamond \end{cases}.$$

By exploiting this last observation and using Lemmas 1 and 2, we provide confidence intervals defined locally at each infoset $J \in \mathcal{J}$ for the realization probabilities $\boldsymbol{y}^*[\sigma_j(J)a]$ of sequences $\sigma_j(J)a$ terminating with an action $a \in A(J)$ at $J$. By letting $\boldsymbol{y}^t \in \mathbb{R}^{|\Sigma_j|}$ be such that:

$$\boldsymbol{y}^t[\sigma_j(J)a] := \frac{N^t(J, a)}{t \bar{\rho}_{-j}^t(J)} \quad \forall J \in \mathcal{J}, \forall a \in A(J), \tag{2}$$

we have the following lemma:

**Lemma 3.** *Let $J \in \mathcal{J}$ be a player $j$'s infoset. Then, for any $0 < \delta \leq 3\exp(-4|A(J)|/5)$ it holds:*

$$\mathbb{P}\left( \bigcap_{a \in A(J)} \left\{ \left| \boldsymbol{y}^t[\sigma_j(J)a] - \boldsymbol{y}^*[\sigma_j(J)a] \right| \leq \frac{5}{2\bar{\rho}_{-j}^t(J)} \sqrt{\frac{\ln(3/\delta)}{t}} \right\} \right) \geq 1 - \delta.$$

Lemma 3 shows that one can use $\boldsymbol{y}^t[\sigma_j(J)a]$ to estimate the realization probability $\boldsymbol{y}^*[\sigma_j(J)a]$ of some sequence $\sigma_j(J)a$, where $\boldsymbol{y}^t[\sigma_j(J)a]$ represents the observed frequency of action $a$ at infoset $J$, adjusted by the average probability of reaching $J$ due to other players, namely $\bar{\rho}^t(J)$. Thus, $\boldsymbol{y}^t$ generalizes the empirical frequency $\boldsymbol{p}^t$ in the single-infoset case to the sequence-form setting. Moreover, Lemma 3 gives high-probability confidence intervals for probabilities $\boldsymbol{y}^*[\sigma_j(J)a]$ at every infoset $J \in \mathcal{J}$, which is used in the following theorem that gives us the desired high-probability confidence region $\mathcal{Y}^t$.

**Theorem 1.** *For every player $j$'s infoset $J \in \mathcal{J}$, let $\delta_J \in (0, 1)$ be such that the condition in Lemma 3 is satisfied and $\sum_{J \in \mathcal{J}} \delta_J < 1$. Then, $\mathbb{P}(\boldsymbol{y}^* \in \mathcal{Y}^t) \geq 1 - \delta$, where $\delta := \sum_{J \in \mathcal{J}} \delta_J$ and*

$$\mathcal{Y}^t := \left\{ \boldsymbol{y} \in \mathcal{Y} : \left| \boldsymbol{y}^t[\sigma_j(J)a] - \boldsymbol{y}^*[\sigma_j(J)a] \right| \leq \frac{5}{2\bar{\rho}_{-j}^t(J)} \sqrt{\frac{\ln(3/\delta_J)}{t}} \quad \forall J \in \mathcal{J}, \forall a \in A(J) \right\}.$$

### 4.2 Constructing the utility-constrained strategy set

We show how to construct the utility-constrained strategy set $\mathcal{X}^t$—for some $\delta \in (0, 1)$ and $\alpha, \beta \in \mathbb{R}$— by exploiting the high-probability confidence region $\mathcal{Y}^{t-1}$ (see Theorem 1). Our approach is to force the condition that each $\boldsymbol{x}$ in the utility-constrained strategy set must satisfy with high probability (see Definition 1 for such condition) for every $\boldsymbol{y} \in \mathcal{Y}^{t-1}$; formally, we require $\alpha \leq \boldsymbol{x}^\top \boldsymbol{U}_j \boldsymbol{y} \leq \beta$ for all $\boldsymbol{y} \in \mathcal{Y}^t$. By definition of $\mathcal{Y}^{t-1}$, this guarantees that such constraint holds also for $\boldsymbol{y}^*$ with probability at least $1 - \delta$. Formally, we define

$$\mathcal{X}^t := \left\{ \boldsymbol{x} \in \mathcal{X} : \max_{\boldsymbol{y} \in \mathcal{Y}^{t-1}} \boldsymbol{x}^\top \boldsymbol{U}_j \boldsymbol{y} \leq \beta \ \wedge \ \min_{\boldsymbol{y} \in \mathcal{Y}^{t-1}} \boldsymbol{x}^\top \boldsymbol{U}_j \boldsymbol{y} \geq \alpha \right\},$$

so that, for any $\boldsymbol{x} \in \mathcal{X}^t$, the upper limit $\beta$ and the lower limit $\alpha$ are satisfied for every $\boldsymbol{y} \in \mathcal{Y}^{t-1}$.

In what follows, we characterize the set $\mathcal{X}^t$ by means of a set of linear inequalities. We formulate the constrained maximization problem, *i.e.*, $\max_{\boldsymbol{y} \in \mathcal{Y}^{t-1}} \boldsymbol{x}^\top \boldsymbol{U}_j \boldsymbol{y}$, using the following linear program:

$$\max_{\boldsymbol{y} \geq \boldsymbol{0}} \boldsymbol{x}^\top \boldsymbol{U}_j \boldsymbol{y} \quad \text{s.t.} \tag{3a}$$

$$\boldsymbol{F}_j \boldsymbol{y} = \boldsymbol{f}_j \tag{3b}$$

$$\boldsymbol{y} \leq \boldsymbol{y}^{t-1} + \boldsymbol{\epsilon}^{t-1} \tag{3c}$$

$$\boldsymbol{y} \geq \boldsymbol{y}^{t-1} - \boldsymbol{\epsilon}^{t-1}, \tag{3d}$$

where we let $\boldsymbol{\epsilon}^{t-1} \in \mathbb{R}_+^{|\Sigma_j|}$ be such that $\boldsymbol{\epsilon}^{t-1}[\sigma_j(J)a] := \frac{5}{2\bar{\rho}_{-j}^t(J)} \sqrt{\frac{\ln(3/\delta_J)}{t}}$ for $J \in \mathcal{J}$ and $a \in A(J)$, for some $\delta_J \in (0,1)$ that satisfy the condition in Lemma 3 and $\sum_{J \in \mathcal{J}} \delta_J \leq \delta$. Notice that, by recalling the definition of $\boldsymbol{y}^{t-1}$ in Equation (2) and that of $\mathcal{Y}^{t-1}$ in Theorem 1, Constraints (3b), (3c), and (3d) correctly encode the fact that $\boldsymbol{y}$ must belong to the set $\mathcal{Y}^{t-1}$.

The dual of Problem (3) reads as: $\min_{\boldsymbol{u} \geq \boldsymbol{0}} \boldsymbol{b}^\top \boldsymbol{u}$ s.t. $\boldsymbol{A}^\top \boldsymbol{u} \geq \boldsymbol{U}_j^\top \boldsymbol{x}$, where $\boldsymbol{u} \in \mathbb{R}^{(|\mathcal{J}|+1) \times 2|\Sigma_j|}$ is a vector of dual variables, while $\boldsymbol{b}$ and $\boldsymbol{A}$ are suitably-defined vector and matrix, respectively. By strong duality, the optimal dual objective equates the optimal primal objective, and, thus, given any player $i$'s strategy $\boldsymbol{x} \in \mathcal{X}$, the condition $\max_{\boldsymbol{y} \in \mathcal{Y}^{t-1}} \boldsymbol{x}^\top \boldsymbol{U}_j \boldsymbol{y} \leq \beta$ holds if there exists a dual feasible solution $\boldsymbol{u}$ that satisfies the additional constraint that $\boldsymbol{b}^\top \boldsymbol{u} \leq \beta$.

By following an analogous reasoning for $\min_{\boldsymbol{y} \in \mathcal{Y}^{t-1}} \boldsymbol{x}^\top \boldsymbol{U}_j \boldsymbol{y}$, and letting $\boldsymbol{\omega} \in \mathbb{R}^{(|\mathcal{J}|+1) \times 2|\Sigma_j|}$ be a vector of variables of the dual problem corresponding to its linear programming formulation (similar to Problem (3)), we state the following main result:

**Theorem 2.** *Let $t \in [T]$ and $\delta \in (0,1)$. Given $\alpha \in \mathbb{R}$ and $\beta \in \mathbb{R}$, it holds:*

$$\mathcal{X}^t = \mathcal{X} \cap \left\{ (\boldsymbol{x}, \boldsymbol{u}, \boldsymbol{\omega}) : \boldsymbol{u}, \boldsymbol{\omega} \geq \boldsymbol{0}, \ \boldsymbol{b}^\top \boldsymbol{u} \leq \beta, \ \boldsymbol{A}^\top \boldsymbol{u} \geq \boldsymbol{U}_j^\top \boldsymbol{x}, \ -\boldsymbol{b}^\top \boldsymbol{\omega} \geq \alpha, \ -\boldsymbol{A}^\top \boldsymbol{\omega} \leq \boldsymbol{U}_j^\top \boldsymbol{x} \right\}.$$

From Theorem 2, it follows that $\mathcal{X}^t$ can be characterized by a polynomially-sized set of linear constraints, which achieves our initial goal. Notice that, in general, the set $\mathcal{X}^t$ defined in Theorem 2 could be empty. This occurs when the linear program used to build $\mathcal{X}^t$ is unfeasible. In practice, this is *not* an issue, since it is always guaranteed to be feasible for $t$ large enough. A complete discussion on this point is deferred to Appendix B.

## 5 How to select the strategy to play

In this section, we provide the implementation of the procedure SELECTSTRATEGY($\mathcal{X}^t, \mathcal{Y}^{t-1}$) in Algorithm 1. To guarantee that COX-UCB attains sub-linear regret after $T$ iterations, *i.e.*, $R^T = o(T)$ (with high probability), we adopt an approach inspired from arm-selection strategies used in *linear* multi-armed bandit problems [Auer, 2002]. In particular, we propose one that uses upper confidence bounds, which is inspired by the LinUCB algorithm [Abbasi-Yadkori *et al.*, 2011].

As a first step, we need to ensure that the algorithm performs enough exploration during game playing. This is crucial to lower bound the probabilities $\bar{\rho}_{-j}^t(J)$ that appear in the bounds defining the set $\mathcal{Y}^t$ (see Theorem 1), and, ultimately, to obtain sub-linear regret. In particular, at each $t \in [T]$, the COX-UCB algorithm selects a strategy $\boldsymbol{x}^t$ that belongs to the following subset of $\mathcal{X}^t$:

$$\tilde{\mathcal{X}}^t := \left\{ \boldsymbol{x} \in \mathcal{X}^t : \boldsymbol{x}[\sigma_i(z)] \geq \alpha^t \ \forall z \in Z \right\},$$

where the $\alpha^t$s are suitably-defined parameters that decrease with the iteration number $t$.[9]

---

[9] Notice that a naïve way of adding exploration to the algorithm would be to use an $\epsilon$-greedy policy that plays the selected strategy $\boldsymbol{x}^t \in \mathcal{X}^t$ with probability $1 - \epsilon$ and a random one with probability $\epsilon$. However, this approach does *not* work in our setting, as it would result in a violation of the constraints on the human's expected utility. On the other hand, picking $\boldsymbol{x} \in \tilde{\mathcal{X}}^t$ assures that such constraints are satisfied.

| **Algorithm 2** Strategy selection of COX-UCB |
|---|

1: **function** SELSTRAT-UCB($\mathcal{X}^t, \mathcal{Y}^{t-1}$)
2:     $\boldsymbol{x}^t \leftarrow \underset{\boldsymbol{x} \in \tilde{\mathcal{X}}^t}{\text{argmax}} \ \underset{\boldsymbol{y} \in \mathcal{Y}^{t-1}}{\max} \ \boldsymbol{x}^\top \boldsymbol{U}_i \boldsymbol{y}$
3:     **return** $\boldsymbol{x}^t$

| **Algorithm 3** Strategy selection of $\psi$-COX-UCB |
|---|

1: **function** SELSTRAT-$\psi$-UCB($\mathcal{X}^t, \mathcal{Y}^{t-1}$)
2:     with probability $1 - \psi$ do:
3:         $\boldsymbol{x}^t \leftarrow \underset{\boldsymbol{x} \in \tilde{\mathcal{X}}^t}{\text{argmax}} \ \underset{\boldsymbol{y} \in \mathcal{Y}^{t-1}}{\max} \ \boldsymbol{x}^\top \boldsymbol{U}_i \boldsymbol{y}$
4:     with probability $\psi$ do:
5:         $\boldsymbol{x}^t \leftarrow \underset{\boldsymbol{x} \in \tilde{\mathcal{X}}^t}{\text{argmax}} \ \boldsymbol{x}^\top \boldsymbol{U}_i \boldsymbol{y}^{t-1}$
6:     **return** $\boldsymbol{x}^t$

The strategy selection mechanism implemented by COX-UCB is provided in Algorithm 2. It is based on the *optimism in face of uncertainty* principle, and, thus, it selects a strategy $\boldsymbol{x}^t \in \tilde{\mathcal{X}}^t$ that maximizes player $i$'s expected payoff $\boldsymbol{x}^\top \boldsymbol{U}_i \boldsymbol{y}$ under the assumption that, for every $\boldsymbol{x} \in \tilde{\mathcal{X}}^t$, strategy $\boldsymbol{y}$ is an optimistic estimate of $\boldsymbol{y}^*$ taken from the confidence region $\mathcal{Y}^{t-1}$, that is, $\boldsymbol{y} \in \mathcal{Y}^{t-1}$ maximizes the same player $i$'s expected payoff $\boldsymbol{x}^\top \boldsymbol{U}_i \boldsymbol{y}$. The following theorem provides a high-probability sub-linear regret guarantee for COX-UCB (its complete formal proof can be found in Appendix C).

**Theorem 3.** *Let* $\alpha^t := \eta \frac{2 \ln^2 t + \ln t + 1}{\sqrt{\ln t} \, (\ln t + 1)^2}$ *for every* $t \in [T]$, *where* $\eta \in (0, 1)$, *and let* $\delta \in (0, 1)$. *The COX-UCB algorithm attains the following regret bound with probability at least* $1 - \delta$:

$$R^T \leq \frac{5}{2\eta} K_{\boldsymbol{U}_i} C \left( 1 + 2\sqrt{T \ln T} \right),$$

*where* $K_{\boldsymbol{U}_i} := \|\boldsymbol{U}_i\|_\infty$ *and* $C$ *is a suitably-defined constant.*

Let us remark that to obtain sub-linear regret only the term $1/\sqrt{\log t}$ is required in the definition of $\alpha^t$; the other terms are added so as to obtain an analytical formula for the regret. Moreover, notice that COX-UCB needs to solve a linearly-constrained *bilinear* optimization problem at each iteration, which can be done efficiently by cutting-hedge solvers (see Section 6).

## 6 Experimental evaluation

We empirically evaluate the convergence of our COX-UCB algorithm on a standard testbed of Kuhn and Leduc poker games [Kuhn, 2016]. We report here the evaluation on rank 7 Kuhn poker (*kuhn_7*) and rank 2 Leduc poker (*leduc_2*), while other experiments and details are deferred to Appendix D.

### 6.1 Approximated version of the COX-UCB algorithm

The COX-UCB algorithm presented in Section 5 (Algorithm 2) solves a bilinear optimization problem at every iteration, with the optimization done over the whole utility-constrained strategy set. Furthermore, let us recall that such set is directly built from the high-confidence region for the human's strategy, which is specified at time $t$ by an estimate $\boldsymbol{y}^t$ and bounds $\boldsymbol{\epsilon}^t$. Empirical evidence from early experiments showed that, despite having the same worst-case convergence rate, in practice the estimate $\boldsymbol{y}^t$ converges much faster to $\boldsymbol{y}^*$ than the bound $\boldsymbol{\epsilon}^t$ does to $\boldsymbol{0}$. We propose a simple modification of COX-UCB, called $\psi$-approximated COX-UCB ($\psi$-COX-UCB), exploiting the faster convergence of the strategy estimation. The only difference between COX-UCB and $\psi$-COX-UCB is in the strategy selection procedure, as specified in Algorithm 3. This simple variation has a twofold effect on the overall algorithm: (i) in some iterations it avoids solving the bilinear program, by solving a much simpler *linear program* instead, and (ii) it leverages the fast empiric convergence of $\boldsymbol{y}^t$ to converge faster. In the remaining part of this section, we empirically evaluate COX-UCB and $\psi$-COX-UCB for different values of the parameter $\psi$.

### 6.2 Experimental results

**Experimental setting.** In order to compare the performances of COX-UCB with those of its approximated version, we consider three versions of $\psi$-COX-UCB, respectively with $\psi = 0.5$, $\psi = 0.7$ and $\psi = 0.9$. We evaluate the performances of the algorithms against 10 different randomly generated strategies for each game instance considered. For each strategy, we execute 5 different algorithm runs. The values $(\alpha, \beta)$ needed for the utility constraints are set to $\alpha = -0.3$ and $\beta = 0.3$ in

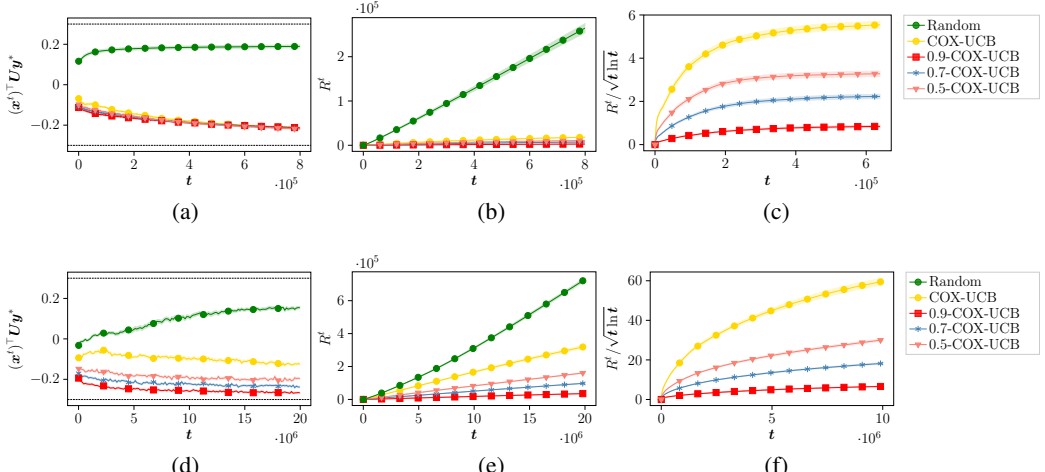

Figure 1: Performances of COX-UCB in Kuhn poker with 5 ranks (first row) and Leduc poker with 2 ranks (second row). From left to right: player $j$'s utility, cumulative regret, and cumulative regret divided by $\sqrt{t \ln t}$.

all the experiments. We use Gurobi for solving bilinear optimization problems [Gurobi Optimization, 2021]. As a baseline for comparisons, we use a *random* algorithm that, at each iteration $t \in [T]$, returns a strategy randomly picked from the utility-constrained strategy set $\mathcal{X}^t$. For further details on the implementation of the algorithm and on the values of the hyper-parameters, see Appendix D.

**Convergence results.** Figure 1 shows the results of the experiments on *kuhn_7* and *leduc_2*. We observe that, in *kuhn_7*, the performances of COX-UCB and those of its approximated versions are comparable, thus showing that the rate of convergence of the opponent's strategy estimation plays a negligible role in this instance. On the other hand, the analysis of *leduc_2* highlights a different behavior, suggesting that, in bigger game instances, the dimension of the confidence bound on the opponent's strategy and the rate with which such dimension reduces become crucial in order to achieve good performances. Furthermore, Figures 1a and 1d show how the algorithm manages to maintain the expected utility of the opponent in the utility range $[-0.3, 0.3]$. On the other hand, we can see how the random algorithm, which plays a random strategy at each iteration, does indeed maintain the utility constraints satisfied (as they are below the threshold of $\beta = 0.3$), but it does *not* result in sublinear regret $R^T$, as one can observe from Figures 1b and 1e. Interestingly, in Figures 1c and 1f the ratio between the regret and $\sqrt{t \ln t}$, representing its asymptotic dependence on $t$, converges to a constant as $t$ increases, thus showing that our upper bound on the regret is tight.

## 7 Conclusions and future directions

In this paper, we studied, for the first time, the problem of designing artificial agents that learn how to exploit an unknown human opponent in an online fashion while, at the same time, guaranteeing that the human's expected utility remains bounded within some upper and lower thresholds. This framework finds application in several real-world human-agent interactions, such as repeated human-agent game playing in which the agent's goal is also to keep the human *engaged* in the repeated interaction, and human teaching by means of *serious games*. Our results hold up under the assumption that the human player adopts a fixed (unknown) stochastic strategy. This is a first crucial step towards more complex models of the human behavior. Nevertheless, even this basic case begets considerable technical challenges, which make the theoretical study of the problem interesting in its own, while also constituting a starting point for the analysis of more complex scenarios.

An interesting direction for future research is assuming a more complex human behavior, such as the case in which the human is learning over time. Another line of research is the enhancement of the scalability of the COX-UCB algorithm, so as to enable its application in real-world sequential games. Finally, it could be of general interest framing our algorithm in serious games scenarios, which have been largely ignored by the research community in artificial intelligence so far.

## Acknowledgments and Disclosure of Funding

This work has been partially supported by the Italian MIUR PRIN 2017 Project ALGADIMAR "Algorithms, Games, and Digital Market".

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
