# Appendix of the paper "Exploiting Opponents under Utility Constraints in Sequential Games"

The appendix is structured as follows:

- Appendix A provides the proofs omitted from Section 4.1, describing the method adopted for the construction of the confidence region $\mathcal{Y}^{t-1}$ for the human strategy $\boldsymbol{y}^*$.
- Appendix B provides the proofs omitted from Section 4.2, describing the method adopted for the construction of the utility-constrained strategy set $\mathcal{X}^t$ starting from $\mathcal{Y}^{t-1}$.
- Appendix C gives the proof omitted from Section 5 for the regret bound of COX-UCB.
- Appendix D provides some additional experimental results.

## A  Proofs omitted from Section 4.1

**Lemma 2.** *Let $\boldsymbol{p} \in \Delta^{|A|}$ and $a^1, \ldots, a^t \in A$ be $t$ actions sampled independently according to $\boldsymbol{p}$. Then, for any $0 < \delta \le 3 \exp\left(-4|A|/5\right)$, it holds:*

$$\mathbb{P}\left( \bigcap_{a \in A} \left\{ \left| \boldsymbol{p}^t[a] - \boldsymbol{p}[a] \right| \le \frac{5}{2} \sqrt{\frac{\ln\left(3/\delta\right)}{t}} \right\} \right) \ge 1 - \delta.$$

*Proof.* Notice that, given the result in Lemma 1, it is sufficient to show that, for every $\epsilon > 0$, it holds

$$\sum_{a \in A} \left| \boldsymbol{p}^t[a] - \boldsymbol{p}[a] \right| \le \epsilon \implies \bigcap_{a \in A} \left\{ \left| \boldsymbol{p}^t[a] - \boldsymbol{p}[a] \right| \le \frac{\epsilon}{2} \right\}.$$

In the following, we prove by contradiction that, if $\left| \boldsymbol{p}^t[a] - \boldsymbol{p}[a] \right| > \frac{\epsilon}{2}$ for some $a \in A$, then $\sum_{a \in A} \left| \boldsymbol{p}^t[a] - \boldsymbol{p}[a] \right| > \epsilon$. Let $\bar{a} \in A$ be such that

$$\epsilon_{\bar{a}} := \boldsymbol{p}^t[\bar{a}] - \boldsymbol{p}[\bar{a}] > \frac{\epsilon}{2}. \tag{4}$$

Then, we have that

$$\sum_{a \in A: a \neq \bar{a}} \boldsymbol{p}^t[a] - \boldsymbol{p}[a] = \sum_{a \in A: a \neq \bar{a}} \boldsymbol{p}^t[a] - \sum_{a \in A: a \neq \bar{a}} \boldsymbol{p}[a] = 1 - \boldsymbol{p}^t[\bar{a}] - 1 + \boldsymbol{p}[\bar{a}] = -\epsilon_{\bar{a}},$$

which, in turn, implies that $\left| \sum_{a \in A: a \neq \bar{a}} \boldsymbol{p}^t[a] - \boldsymbol{p}[a] \right| = \epsilon_{\bar{a}} > \left| \frac{\epsilon}{2} \right|$. Moreover, by the triangular inequality, we have:

$$\sum_{a \in A: a \neq \bar{a}} \left| \boldsymbol{p}^t[a] - \boldsymbol{p}[a] \right| \ge \left| \sum_{a \in A: a \neq \bar{a}} \boldsymbol{p}^t[a] - \boldsymbol{p}[a] \right| > \epsilon/2. \tag{5}$$

By summing Equation (4) and Equation (5), we obtain $\sum_{a \in A} \left| \boldsymbol{p}^t[a] - \boldsymbol{p}[a] \right| > \epsilon$, which is the desired contradiction that proves the result. $\square$

**Lemma 3.** *Let $J \in \mathcal{J}$ be a player $j$'s infoset. Then, for any $0 < \delta \le 3 \exp(-4|A(J)|/5)$ it holds:*

$$\mathbb{P}\left( \bigcap_{a \in A(J)} \left\{ \left| \boldsymbol{y}^t[\sigma_j(J)a] - \boldsymbol{y}^*[\sigma_j(J)a] \right| \le \frac{5}{2\bar{\rho}_{-j}^t(J)} \sqrt{\frac{\ln\left(3/\delta\right)}{t}} \right\} \right) \ge 1 - \delta.$$

*Proof.* Since $\boldsymbol{n}_J^t$ follows a multinomial distribution, using Lemma 1 provides us with an high-probability confidence region for the components of $\boldsymbol{y}^*$ corresponding to the sequences terminating with an action at infoset $J$. Formally, since $\boldsymbol{p}^t$ in Lemma 1 plays the same role as $\frac{1}{t} \boldsymbol{n}_J^t$, we get:

$$\mathbb{P}\left( \sum_{a \in A(J)} \left| N^t(J, a) - \mathbb{E}[N^t(J, a)] \right| \le 5t \sqrt{\frac{\ln(3/\delta)}{t}} \right) \ge 1 - \delta.$$

Let us recall that $\mathbb{E}[N^t(J,a)] = t\,\bar{\rho}^t_{-j}(J)\boldsymbol{y}^*[\sigma_j(J)a]$. Thus, dividing by $t\,\bar{\rho}^t_{-j}(J)$ the argument of the probability in the left hand side of the above equation, we get:

$$\mathbb{P}\left(\sum_{a\in A(J)}\left|\frac{N^t(J,a)}{t\,\bar{\rho}^t_{-j}(J)} - \boldsymbol{y}^*[\sigma_j(J)a]\right| \le \frac{5}{\bar{\rho}^t_{-j}(J)}\sqrt{\frac{\ln(3/\delta)}{t}}\right) \ge 1-\delta. \tag{6}$$

Following the same line of reasoning of the proof of Lemma 2, we conclude that:

$$\mathbb{P}\left(\bigcap_{a\in A(J)}\left\{\left|\frac{N^t(J,a)}{t\bar{\rho}^t_{-j}(J)} - \boldsymbol{y}^*[\sigma_j(J)a]\right| \le \frac{5}{2\bar{\rho}^t_{-j}(J)}\sqrt{\frac{\ln(3/\delta)}{t}}\right\}\cap E_\diamond\right) \ge 1-\delta, \tag{7}$$

where we define the event $E_\diamond := \left\{\left|N^t(J,a_\diamond) - \mathbb{E}[N^t(J,a_\diamond)]\right| \le \frac{5t}{2}\sqrt{\frac{\ln(3/\delta)}{t}}\right\}$. The statement follows from the fact that, for two generic events $E$ and $E'$, it holds $\mathbb{P}(E\cap E') \le \mathbb{P}(E)$. $\square$

**Theorem 1.** *For every player $j$'s infoset $J \in \mathcal{J}$, let $\delta_J \in (0,1)$ be such that the condition in Lemma 3 is satisfied and $\sum_{J\in\mathcal{J}}\delta_J < 1$. Then, $\mathbb{P}(\boldsymbol{y}^* \in \mathcal{Y}^t) \ge 1-\delta$, where $\delta := \sum_{J\in\mathcal{J}}\delta_J$ and*

$$\mathcal{Y}^t := \left\{\boldsymbol{y}\in\mathcal{Y} : \left|\boldsymbol{y}^t[\sigma_j(J)a] - \boldsymbol{y}^*[\sigma_j(J)a]\right| \le \frac{5}{2\bar{\rho}^t_{-j}(J)}\sqrt{\frac{\ln(3/\delta_J)}{t}} \quad \forall J\in\mathcal{J}, \forall a\in A(J)\right\}.$$

*Proof.* For each infoset $J \in \mathcal{J}$, let us apply Lemma 3 with $\delta = \delta_J \le 3\exp(-4|A(J)|/5)$. The lemma states that for each $J \in \mathcal{J}$, the event

$$E_J := \bigcap_{a\in A(J)}\left\{\left|\boldsymbol{y}^t[\sigma_j(J)a] - \boldsymbol{y}^*[\sigma_j(J)a]\right| \le \frac{5}{2\bar{\rho}^t_{-j}(J)}\sqrt{\frac{\ln(3/\delta_J)}{t}}\right\} \tag{8}$$

holds with probability at least $1-\delta_J$. By applying a union bound, we have that:

$$\mathbb{P}\left(\bigcap_{J\in\mathcal{J}}E_J\right) = 1 - \mathbb{P}\left(\bigcup_{J\in\mathcal{J}}\neg E_J\right)$$

$$\ge 1 - \sum_{J\in\mathcal{J}}1 - \mathbb{P}(E_J)$$

$$\ge 1 - \sum_{J\in\mathcal{J}}\delta_J$$

$$= 1 - \delta.$$

Finally, choosing the errors $\delta_J$ such that $\sum_{J\in\mathcal{J}}\delta_J < 1$ proves the result. $\square$

## B    Proofs omitted from Section 4.2

**Theorem 2.** *Let $t \in [T]$ and $\delta \in (0,1)$. Given $\alpha \in \mathbb{R}$ and $\beta \in \mathbb{R}$, it holds:*

$$\mathcal{X}^t = \mathcal{X} \cap \left\{(\boldsymbol{x},\boldsymbol{u},\boldsymbol{\omega}) : \boldsymbol{u},\boldsymbol{\omega}\ge\boldsymbol{0},\ \boldsymbol{b}^\top\boldsymbol{u}\le\beta,\ \boldsymbol{A}^\top\boldsymbol{u}\ge\boldsymbol{U}_j^\top\boldsymbol{x},\ -\boldsymbol{b}^\top\boldsymbol{\omega}\ge\alpha,\ -\boldsymbol{A}^\top\boldsymbol{\omega}\le\boldsymbol{U}_j^\top\boldsymbol{x}\right\}.$$

*Proof.* The proof follows the reasoning outlined in Section 4.2. First, we notice that $\boldsymbol{x}\in\mathcal{X}$ belongs to the utility-constrained strategy set $\mathcal{X}^t$ at iteration $t\in[T]$ if and only if

$$\max_{\boldsymbol{y}\in\mathcal{Y}^{t-1}}\boldsymbol{x}^\top\boldsymbol{U}_j\boldsymbol{y}\le\beta \ \wedge\ \min_{\boldsymbol{y}\in\mathcal{Y}^{t-1}}\boldsymbol{x}^\top\boldsymbol{U}_j\boldsymbol{y}\ge\alpha.$$

By first considering the max problem, we can write it as the linear program in Problem (3) in the main paper. Then, its dual problem reads as follows:

$$\min_{\boldsymbol{u}\ge\boldsymbol{0}}\boldsymbol{b}^\top\boldsymbol{u} \quad \text{s.t.} \tag{9a}$$

$$\boldsymbol{A}^\top\boldsymbol{u}\ge\boldsymbol{U}_j^\top\boldsymbol{x}, \tag{9b}$$

where $\boldsymbol{u} \in \mathbb{R}^{(|\mathcal{J}|+1) \times 2|\Sigma_j|}$ is a vector of dual variables, while:

$$\boldsymbol{A} := \begin{bmatrix} \boldsymbol{I}_{|\Sigma_j|} \\ -\boldsymbol{I}_{|\Sigma_j|} \\ \boldsymbol{F}_j \\ -\boldsymbol{F}_j \end{bmatrix} \quad \text{and} \quad \boldsymbol{b} := \begin{bmatrix} \boldsymbol{y}^{t-1} + \boldsymbol{\epsilon}^{t-1} \\ -\boldsymbol{y}^{t-1} + \boldsymbol{\epsilon}^{t-1} \\ \boldsymbol{f}_j \\ -\boldsymbol{f}_j \end{bmatrix},$$

with $\boldsymbol{I}_n$ being the $n \times n$ identity matrix. By strong duality, the optimal dual objective equates the optimal primal objective, and, thus, given any player $i$'s strategy $\boldsymbol{x} \in \mathcal{X}$, the condition $\max_{\boldsymbol{y} \in \mathcal{Y}^{t-1}} \boldsymbol{x}^\top \boldsymbol{U}_j \boldsymbol{y} \leq \beta$ holds if there exists a dual feasible solution $\boldsymbol{u}$ that satisfies the additional constraint that $\boldsymbol{b}^\top \boldsymbol{u} \leq \beta$. Following an analogous reasoning for $\min_{\boldsymbol{y} \in \mathcal{Y}^{t-1}} \boldsymbol{x}^\top \boldsymbol{U}_j \boldsymbol{y}$, and letting $\boldsymbol{\omega} \in \mathbb{R}^{(|\mathcal{J}|+1) \times 2|\Sigma_j|}$ be a vector of variables of the dual problem corresponding to its linear programming formulation (similar to Problem (3)), the result follows. $\square$

**Discussion on the emptiness of $\mathcal{X}^t$ in Theorem 2.** In some cases, the set $\mathcal{X}^t$ defined in Theorem 2 could be empty. This happens when the components of the vector $\boldsymbol{\epsilon}^t$ are large, as it is usually the case after the first game repetitions. However, since by assumption the problem is feasible for the true sequence-form strategy $\boldsymbol{y}^*$, then the set $\mathcal{X}^t$ will be non-empty after a finite number of iterations. Hence, as customary in safe exploration problems, we can assume that we have at our disposal an initial number of plays that allow to have an estimate of $\boldsymbol{y}^*$ that is good enough (*i.e.*, with a small norm of $\boldsymbol{\epsilon}^t$) so that $\mathcal{X}^t$ is non-empty. In practice, one does *not* need to wait that $\mathcal{X}^t$ is always non-empty, and can mix the initial pure-exploration phase with the selection strategy implemented by the algorithm. For instance, this can be achieved by playing a random strategy when $\mathcal{X}^t$ is empty, while following the algorithm recommendation when $\mathcal{X}^t$ is non-empty. In the experimental evaluation, which is discussed in details in Appendix D, we use the variable `N_BLANK_GAMES` to tune this aspect of the algorithm implementation.

## C  Proofs omitted from Section 5

Before proving Theorem 3, we need to show the following technical lemma.

**Lemma 4.** *Let* $f(\tau) := \frac{2 \ln^2 \tau + \ln \tau + 1}{\sqrt{\ln \tau} (\ln \tau + 1)^2}$. *Then, it holds that:*

$$\sum_{\tau=1}^{t} f(\tau) \geq \frac{2t \sqrt{\ln t}}{\ln t + 1}. \tag{10}$$

*Proof.* By noticing that $f(\tau)$ is decreasing in $\tau$, we can use the following integral inequality:

$$\sum_{\tau=1}^{t} f(\tau) \geq \sum_{\tau=1}^{t} \int_{\tau}^{\tau+1} f(x)dx = \int_{1}^{t+1} f(x)dx \geq \int_{1}^{t} f(x)dx = \frac{2t\sqrt{\ln t}}{\ln t + 1},$$

which shows the result. $\square$

**Theorem 3.** *Let* $\alpha^t := \eta \frac{2 \ln^2 t + \ln t + 1}{\sqrt{\ln t} (\ln t + 1)^2}$ *for every* $t \in [T]$, *where* $\eta \in (0, 1)$, *and let* $\delta \in (0, 1)$. *The COX-UCB algorithm attains the following regret bound with probability at least* $1 - \delta$:

$$R^T \leq \frac{5}{2\eta} K_{\boldsymbol{U}_i} C \left(1 + 2\sqrt{T \ln T}\right),$$

*where* $K_{\boldsymbol{U}_i} := ||\boldsymbol{U}_i||_\infty$ *and* $C$ *is a suitably-defined constant.*

*Proof.* First, let us recall that the confidence regions $\mathcal{Y}^{t-1}$ used by the COX-UCB algorithm are built by applying Theorem 1 with error tolerances $\delta_J \in (0, 1)$, for $J \in \mathcal{J}$, such that the conditions in Theorem 1 are satisfied and $\delta = \sum_{J \in \mathcal{J}} \delta_J$. In the following, we prove the desired regret bound by bounding the regret that player $i$ suffers ar each iteration $t$.

For every $t \in [T]$, we let $\boldsymbol{x}^{t,*} \in \operatorname{argmax}_{\boldsymbol{x}^* \in \mathcal{X}^t} (\boldsymbol{x}^*)^\top \boldsymbol{U}_i \boldsymbol{y}^*$. Then, at each iteration $t$, player $i$ incurs in an instantaneous regret $r^t$, which is formally defined as follows:

$$r^t := (\boldsymbol{x}^{t,*})^\top \boldsymbol{U}_i \boldsymbol{y}^* - (\boldsymbol{x}^t)^\top \boldsymbol{U}_i \boldsymbol{y}^*.$$

Since the COX-UCB algorithm selects strategies $\boldsymbol{x}^t$ so that $\boldsymbol{x}^t \in \operatorname{argmax}_{\boldsymbol{x} \in \tilde{\mathcal{X}}^t} \max_{\boldsymbol{y} \in \mathcal{Y}^{t-1}} \boldsymbol{x}^\top \boldsymbol{U}_i \boldsymbol{y}$ (see Algorithm 2), we have that, with probability at least $1 - \delta$, it holds

$$r_t \leq (\boldsymbol{x}^t)^\top \boldsymbol{U}_i \tilde{\boldsymbol{y}}^t - (\boldsymbol{x}^t)^\top \boldsymbol{U}_i \boldsymbol{y}^*, \tag{11}$$

where we let $\tilde{\boldsymbol{y}}^t \in \operatorname{argmax}_{\boldsymbol{y} \in \mathcal{Y}^{t-1}} (\boldsymbol{x}^t)^\top \boldsymbol{U}_i \boldsymbol{y}$. By using the definition of the sequence-form utility matrix $\boldsymbol{U}_i$, we can re-write Equation (11) as follows:

$$r_t \leq (\boldsymbol{x}^t)^\top \boldsymbol{U}_i (\tilde{\boldsymbol{y}}^t - \boldsymbol{y}^*) = \sum_{z \in Z} \boldsymbol{x}^t[\sigma_i(z)] \, \boldsymbol{U}_i[\sigma_i(z), \sigma_j(z)] \left( \tilde{\boldsymbol{y}}^t[\sigma_j(z)] - \boldsymbol{y}^*[\sigma_j(z)] \right).$$

For every terminal node $z \in Z$, by letting $J(z) \in \mathcal{J}$ be the (unique w.l.o.g.) infoset such that the last action of the sequence $\sigma_j(z)$ is played at $J(z)$, we can invoke Theorem 1 together with the Cauchy-Swartz inequality to obtain that, with probability at least $1 - \delta_{J(z)}$, the following holds

$$\tilde{\boldsymbol{y}}^t[\sigma_j(z)] - \boldsymbol{y}^*[\sigma_j(z)] \leq \frac{5}{\bar{\rho}_{-j}^t(J(z))} \sqrt{\frac{\ln(3/\delta_{J(z)})}{t}}. \tag{12}$$

Moreover, since by definition of $\tilde{\mathcal{X}}^t$, we have that $\boldsymbol{x}^t[\sigma_i] \geq \alpha^t$ for all $\sigma_i \in \Sigma_i$. This gives us the following lower bound on the probability $\rho_{-j}^t(J(z))$:

$$\rho_{-j}^t(J(z)) = \sum_{h \in J(z)} \boldsymbol{x}^t[\sigma_i(h)] p_c(h) \geq \alpha^t \sum_{h \in J(z)} p_c(h) = p_c(J(z))\alpha^t, \tag{13}$$

where we let $p_c(J(z)) := \sum_{h \in J(z)} p_c(h)$.

Then, the term $\frac{1}{\bar{\rho}_{-j}^t(J_z)\sqrt{t}}$ in Equation (12) can be bounded as follows.

$$\frac{1}{\bar{\rho}_{-j}^t(J(z))\sqrt{t}} = \frac{\sqrt{t}}{\sum\limits_{\tau=1}^{t} \rho_{-j}^t(J(z))} \leq \frac{\sqrt{t}}{p_c(J(z)) \sum\limits_{\tau=1}^{t} \alpha^\tau} \leq \frac{1}{2\eta p_c(J(z))} \frac{\ln t + 1}{\sqrt{t \ln t}}, \tag{14}$$

where the first inequality follows from Equation (13) and the second one comes from Lemma 4.

Therefore, by combining Equation 12 and Equation (14), we obtain:

$$\tilde{\boldsymbol{y}}^t[\sigma_j(z)] - \boldsymbol{y}^*[\sigma_j(z)] \leq \frac{5\sqrt{\ln(3/\delta_{J(z)})}}{2\eta p_c(J(z))} \frac{\ln t + 1}{\sqrt{t \ln t}}, \tag{15}$$

which holds with probability at least $1 - \delta_{J(z)}$.

Using Equation (15) and observing that $\boldsymbol{x}^t[\sigma_i(z)] \leq 1$ for all $z \in Z$, we can conclude that, with probability at least $1 - \delta$, it holds:

$$R_T := \sum_{t=1}^{T} r_t \leq \sum_{t=1}^{T} (\boldsymbol{x}^t)^\top \boldsymbol{U}_i (\tilde{\boldsymbol{y}}^t - \boldsymbol{y}^*) \tag{16}$$

$$\leq K_{\boldsymbol{U}_i} \sum_{t=1}^{T} \sum_{z \in Z} \frac{5\sqrt{\ln(3/\delta_{J(z)})}}{2\eta p_c(J(z))} \frac{\ln t + 1}{\sqrt{t \ln t}} \tag{17}$$

$$= K_{\boldsymbol{U}_i} \frac{5}{2\eta} \left( \sum_{z \in Z} \frac{\sqrt{\ln(3/\delta_{J(z)})}}{p_c(J(z))} \right) \sum_{t=1}^{T} \frac{\ln t + 1}{\sqrt{t \ln t}} \tag{18}$$

$$\leq K_{\boldsymbol{U}_i} \frac{5}{2\eta} \left( \sum_{z \in Z} \frac{\sqrt{\ln(3/\delta_{J(z)})}}{p_c(J(z))} \right) \left( 1 + \sum_{t=1}^{T-1} \int_{\tau=t}^{t+1} \frac{\ln \tau + 1}{\sqrt{\tau \ln \tau}} d\tau \right) \tag{19}$$

$$= K_{\boldsymbol{U}_i} \frac{5}{2\eta} \left( \sum_{z \in Z} \frac{\sqrt{\ln(3/\delta_{J(z)})}}{p_c(J(z))} \right) \left( 1 + \int_{2}^{T} \frac{\ln \tau + 1}{\sqrt{\tau \ln \tau}} d\tau \right) \tag{20}$$

$$\leq K_{\boldsymbol{U}_i} \frac{5}{2\eta} \left( \sum_{z \in Z} \frac{\sqrt{\ln(3/\delta_{J(z)})}}{p_c(J(z))} \right) \left( 1 + 2\sqrt{T \ln T} \right), \tag{21}$$

where we let $K_{\boldsymbol{U}_i} := \|\boldsymbol{U}_i\|_\infty$ and $C := \sum_{z \in Z} \frac{\sqrt{\ln(3/\delta_{J(z)})}}{p_c(J(z))}$. This concludes the proof. $\qquad \square$

Theorem 3 gives a sublinear upper bound on the regret of the COX-UCB algorithm. Notice that the order of the regret is $\sqrt{T \ln T}$ and that the constant $C$ is linear in the number of terminal nodes $Z$.

# D  Additional details on the experimental evaluation

In this section, we provide additional details and results on the experimental evaluation of our COX-UCB and $\psi$-COX-UCB algorithms. We test the two algorithms in three different instances of Kuhn poker with ranks 3, 5 and 7 (denoted respectively as *kuhn_3*, *kuhn_5* and *kuhn_7*) and in one instance of Leduc poker with ranks 2 (denoted as *leduc_2*).

## D.1  Experimental setting and hyperparameters

In order to guarantee that the utility-constrained strategy set $\mathcal{X}^t$ is non-empty at the beginning of the repeated interaction, we assume to have access to some prior information on the strategy employed by the human player, so as to reduce the initial uncertainty encoded by the confidence region $\mathcal{Y}^{t-1}$. This is reasonable in practice, since a new player can always be profiled according to a number of user classes. In particular, we encode this information as observations collected during a number N_BLANK_GAMES of games played by the players at the beginning of the repeated interaction, in which the agent player adopts a purely-explorative strategy.

Furthermore, since the estimation and bounds do not change significantly after a single game, we compute the solution to the optimization problem required by COX-UCB every UPDATE_EVERY iterations. Moreover, we set a time limit (TIME_LIMIT) to the Gurobi solver to solve the bilinear program. This allows us to reduce significantly the time spent to solve bilinear optimization problems.

In all our experiments, the values of the hyperparameters are set to:

- $\delta = 0.05$ and $\delta_J = \delta/|\mathcal{J}|$ for all $J \in \mathcal{J}$;
- N_BLANK_GAMES $= 1000$;
- UPDATE_EVERY $= 20$;
- TIME_LIMIT $= 1s$;
- $\eta = 0.05/|\Sigma_i|$;
- utility constraints lower bound $\alpha = -0.3$;
- utility constraints upper bound $\beta = 0.3$.

We fix the number of iterations after which we stop the execution of our algorithms to $2e5$, $4e5$, $8e5$ and $2e6$ for *kuhn_3*, *kuhn_5*, *kuhn_7* and *leduc_2*, respectively.

Finally, the infrastructure used to run the experiments is a 32-core UNIX system with 128 GB RAM.

## D.2  Detailed experimental results

Figure 2 shows the performances of COX-UCB and $\psi$-COX-UCB. The values tested for the hyperparameter $\psi$ are $\psi = 0.5$, $\psi = 0.7$ and $\psi = 0.9$. As a baseline we use a random policy that consists in randomly selecting a sequence-form strategy from the set $\mathcal{X}^t$ at every time step $t$. The first column of Figure 2 shows the expected utility of the opponent over the iterations of the algorithm. As we can observe, empirically, the random policy satisfies the utility constraints. This is reasonable, since the strategies are selected from the interior of the utility-constrained strategy set $\mathcal{X}^t$. However, in all the game instances considered, only COX-UCB and $\psi$-COX-UCB approach the optimal utility values, which are shown in Table 1. Looking at the plots of the cumulative regret (second column of Figure 2), we can observe that COX-UCB and $\psi$-COX-UCB achieve a significantly lower regret than the baseline. The experiments on *leduc_2* remark the relevance of the convergence rate of the confidence bound on the opponent's strategy. In particular, when the strategy space is larger—as it is the case of *leduc_2*—, the fact that the confidence bound reduces slowly causes a decrease in the performances, slowing down the convergence to an optimal strategy. In this scenario, the approximation yielded by $\psi$-COX-UCB allows the algorithm to exploit the faster empirical convergence rate of the average strategy, thus resulting in a lower cumulative regret. Finally, the plots on the third column of Figure 2 allow us to evaluate the upper bound derived for the cumulative regret. In particular, we point out

that the ratio between the cumulative regret and $\sqrt{t\ln(t)}$ converges to an horizontal line, meaning that the bounds that we derived are tight.

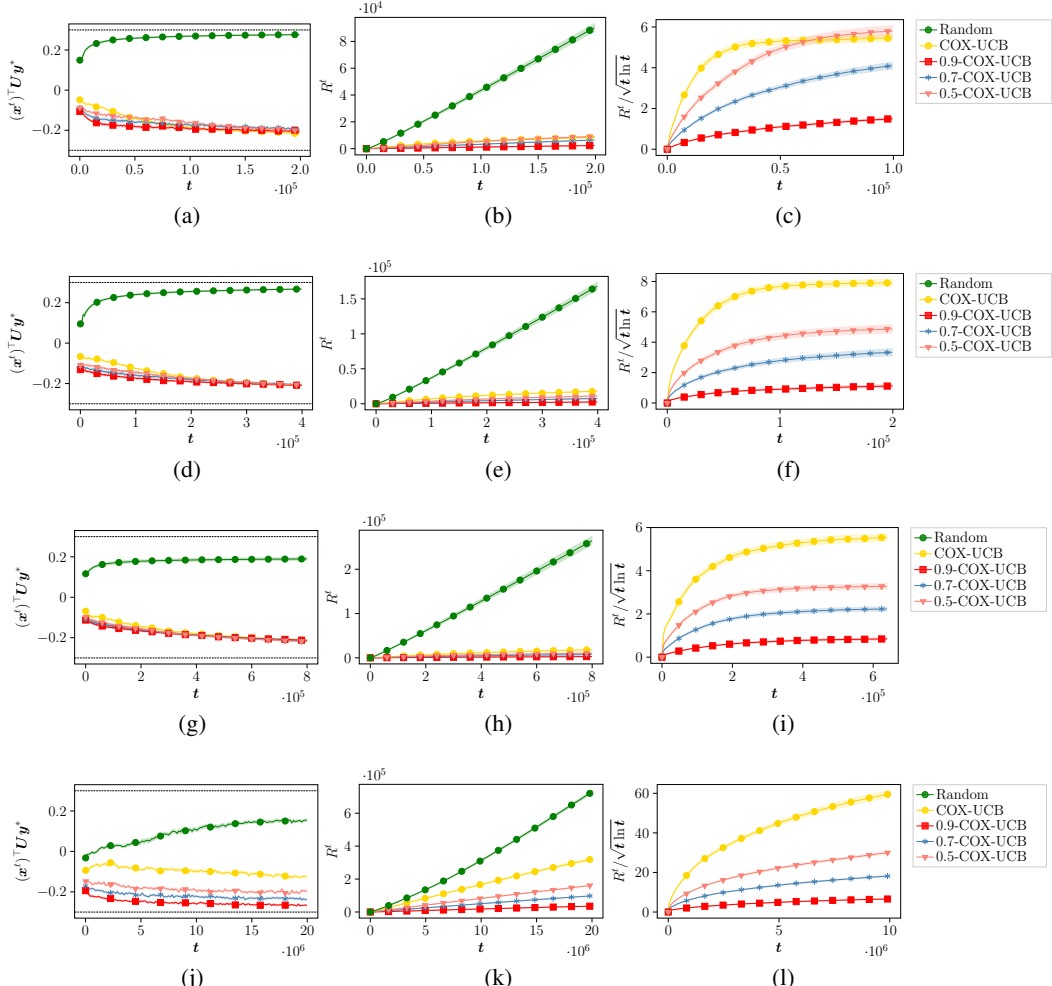

Figure 2: Performances of COX-UCB in Kuhn poker with 3 (top row), 5 (second row) and 7 (third row) ranks and in Leduc poker with 2 ranks (bottom row). From left to right: player $j$'s utility, cumulative regret, and cumulative regret divided by $\sqrt{t\ln t}$.

Table 1: Optimal expected utility for the opponent in *kuhn_3*, *kuhn_5*, *kuhn_7*, and *leduc_2*.

| - | $-\max_{\boldsymbol{x}\in\mathcal{X}^\star} \boldsymbol{x}^\top \boldsymbol{U}_i \boldsymbol{y}^\star$ |
|---|---|
| *kuhn_3* | $-0.28$ |
| *kuhn_5* | $-0.3$ |
| *kuhn_7* | $-0.29$ |
| *leduc_2* | $-0.3$ |

Table 2: Average time per iteration for the algorithms COX-UCB, 0.5-COX-UCB, 0.7-COX-UCB and 0.9-COX-UCB in *kuhn_3*, *kuhn_5*, *kuhn_7*, and *leduc_2*.

| - | COX-UCB | 0.5-COX-UCB | 0.7-COX-UCB | 0.9-COX-UCB |
|---|---|---|---|---|
| *kuhn_3* | $0.011s$ | $0.006s$ | $0.004s$ | $0.003s$ |
| *kuhn_5* | $0.011s$ | $0.006s$ | $0.004s$ | $0.004s$ |
| *kuhn_7* | $0.012s$ | $0.007s$ | $0.005s$ | $0.005s$ |
| *leduc_2* | $0.016s$ | $0.009s$ | $0.007s$ | $0.006s$ |