# OpenReview forum: "Exploiting Opponents Under Utility Constraints in Sequential Games"
_NeurIPS.cc/2021/Conference — NeurIPS 2021 Poster_

### Official Review · Reviewer_26mD · 2021-07-12

**Rating:** 6
**Confidence:** 4

**Summary:**

The paper describes a novel approach for exploiting an unknown human opponent while ensuring that the human's expected payoff remains within a certain interval, thereby keeping them "engaged." The setting is for 2-player zero-sum imperfect-information games where the public actions and private information of the human opponent agent are observed after each round.

**Limitations And Societal Impact:**

I don't really see any major limitations or potential negative societal impact.
The only issue I can think of is that developers could use the algorithm to develop bots that play against humans illegally
on gaming websites that can avoid being easily detected.

**Main Review:**

I think the paper is pretty good overall. I have two major criticisms, and some minor comments.

First, I don’t think the paper does a good job motivating the problem. What is a setting where we want to create an AI agent that can exploit a human opponent while still keeping the human “engaged”? The motivation seems very abstract and hypothetical. Why not just let the human play against other humans? Maybe some humans want to play against a very strong opponent? Why “exploit” the human at all, and why not just play a static strategy that is around the same level as the human? The paper needs to do a much better job motivating why this problem is interesting and provide at least one concrete example problem.

I could think of one motivating scenario, though it is likely not what the authors had in mind. If one wanted to develop a poker bot to play online against humans (which is against the rules of the sites), then on the one hand they would like the bot to win as much as possible, but they do not want the humans and/or the site to detect that they are using a bot. So they want to ensure that they do not win so much as to arouse suspicion.

The second major issue is that the paper makes the significant assumption that the entire sequence of the opponent’s actions is observed. This is very unrealistic in imperfect-information games. For example, in poker, often the public actions of the opponents are observed, but not their private cards. (In some cases the hand goes to a “showdown” and opponent’s private cards are revealed, but frequently there is no showdown and they are not revealed). The authors are implicitly assuming that the opponents’ private information is observed at each iteration, since otherwise we would not what the sequence of the opponent’s information sets and actions are. If the authors want to use this model, they need to be honest about its limitations and be clear that it does not apply to typical imperfect-information games.



15: be -> are
29: quote outside period
29: More in general -> More generally
29: “humans are interested in repeatedly playing against an (artificial or human) opponent when they are sufficiently engaged in the competition” – citation for this?
64: Citation Ganzfried & Sun ’18 doesn’t provide theoretical guarantees for safe opponent exploitation. It presents an exact opponent modeling approach for the setting with a Dirichlet prior and observed public actions but unobserved private information.

Seems to be a contradiction:
70: “We do not make any assumption on the human strategy”
Appendix E.q: “In order to guarantee that the engaging set is non-empty at the beginning of the repeated interaction, we assume to have access to some prior information on the strategy employed by the human player, so as to reduce the initial uncertainty encoded by the confidence region.”
What exactly is assumed about the human strategy?

74: merit to be investigated -> merit investigation
76: Notice that, -> Notice that
Footnote 1: empiric -> empirical
Footnote 1: “However, its empiric performance being very poor, we” -> “However, since its empirical performance is very poor, we:
179: observed up the -> observed up to the
310: cutting-hedge -> cutting-edge
311: empiric -> empirical
The rules of Kuhn poker are never described, not even in the appendix.
316: levels--, and -> levels—

In the experiments, it would be interesting to include a static Nash equilibrium strategy as a benchmark, and to see whether the opponent’s payoff falls within the “engagement” threshold range for it. It would also be interesting to include a full best response strategy to see how far outside of the engagement range the payoff is.

Is there any precedent for the selection of the opponent strategies by using the ranges of entropy? If so, are there references for other works that use this approach? Or if not, can you elaborate on why this selection is interesting?

351: startegy -> strategy


**Time Spent Reviewing:**

5

---

> ### Author Response · Authors · 2021-08-10
> **Authors’ answer to reviewer 26mD.**
>
> Re: *“First, I don’t think the paper does a good job motivating the problem. What is a setting where we want to create an AI agent that can exploit a human opponent while still keeping the human “engaged”? The motivation seems very abstract and hypothetical. Why not just let the human play against other humans? Maybe some humans want to play against a very strong opponent? Why “exploit” the human at all, and why not just play a static strategy that is around the same level as the human? The paper needs to do a much better job motivating why this problem is interesting and provide at least one concrete example problem.”*:
> We agree with the Reviewer that the paper should motivate the problem better. We spent large efforts on the derivation of the theoretical results and we did not do a good job in motivating the problem, we are really sorry for that. However, we are confident that we can provide better motivations in the camera ready. In particular, the problem of engagement is significant, as also noted by Reviewer zQhR, in each context in which one cares about extending the hours of play of the players, and has interesting motivating applications in serious games (and many other game settings). Our model (with support from psychological literature, see for example Believability Through Psychosocial Behaviour: Creating Bots That Are More Engaging and Entertaining. Believable Bots 2012: 29-68 by Bailey et al., we also cited in answering Reviewer 1aGH) assumes that a player remains engaged in the game when the game is neither too easy nor too difficult, i.e. he/she loses sometimes, but not too much. One can also imagine a scenario in which the thresholds used for the engagement constraints change over time to capture a progression in the utility expected by the human (this does not affect the regret bound of our algorithm). Most importantly, the entire framework also works in general-sum games.
>
> Re: *“The second major issue is that the paper makes the significant assumption that the entire sequence of the opponent’s actions is observed. This is very unrealistic in imperfect-information games. For example, in poker, often the public actions of the opponents are observed, but not their private cards. (In some cases the hand goes to a “showdown” and opponent’s private cards are revealed, but frequently there is no showdown and they are not revealed). The authors are implicitly assuming that the opponents’ private information is observed at each iteration, since otherwise we would not what the sequence of the opponent’s information sets and actions are. If the authors want to use this model, they need to be honest about its limitations and be clear that it does not apply to typical imperfect-information games.”*:
> Thanks for the observation. We run our experiments on Poker, since it is a common test-bed for extensive-form games (however, we agree that we need to specify the remark of the Reviewer in the paper and we will do so). In applications like serious games, however, it is often the case that the agent observes all the opponent’s actions. Moreover the problem of opponent modelling in the presence of private information which is not revealed at the end of the game is discussed in Ganzfried, Sam, and Qingyun Sun. "Bayesian opponent exploitation in imperfect-information games." 2018 IEEE Conference on Computational Intelligence and Games (CIG). IEEE, 2018, and can be used directly to extend our result. We will add this comment in the final version of the paper.
>
> Re: *“What exactly is assumed about the human strategy?”*:
> We do not assume any particular structure or property on the human strategy. In the experiments, in order to achieve a reasonable estimate, we perform N_BLANK_GAMES (see Appendix E.1) warm-up games in which the bot adopts a purely explorative strategy (i.e., it plays randomly at every information set). This practice allows our algorithm to obtain some preliminary knowledge on the human strategy. Notice that without any prior information on the player’s strategy, the bot’s strategy space satisfying the engagement constraints might be empty. So, a prior is needed in practice.
>
> Re: *“Is there any precedent for the selection of the opponent strategies by using the ranges of entropy? If so, are there references for other works that use this approach? Or if not, can you elaborate on why this selection is interesting?”*:
> Dividing the strategies in levels of entropy seemed natural to us since different levels of entropy guarantee different levels of exploration of the game tree, e.g. infosets in which the human player plays only close to pure strategies makes it hard to observe entire sections of the tree. We note that, in the experimental campaign, engaging players with lower entropy was easier than that for players using strategies with high entropy. However, we agree that this consideration should be specified better in the paper.
>
> Re: *“In the experiments, it would be interesting to include a static Nash equilibrium strategy as a benchmark, and to see whether the opponent’s payoff falls within the “engagement” threshold range for it. It would also be interesting to include a full best response strategy to see how far outside of the engagement range the payoff is”*:
> We thank the reviewer for this suggestion on further experiments, we will add such in the final version of the paper. Note that a static Nash Equilibrium does indeed guarantee engagement constraints. However, it results more inefficient than our algorithm as the Nash Equilibrium does not fully exploit the opponent. In particular, in Kuhn5, it guarantees on average an expected utility to the human of ~-0.14 against strategies with levels of entropy 1 and 2 and ~-0.08 against strategies with third level of entropy. Furthermore, the expected utilities of the human against a pure exploitative strategy (best response) in kuhn 5 are on average ~-0.51, ~-0.5, ~-0.4 for the three levels of entropy respectively, thus showing that pure exploitation of the opponent could lead to an uncontrolled loss for the human (beyond the engagement constraints we adopted).

---

### Official Review · Reviewer_1aGH · 2021-07-15

**Rating:** 6
**Confidence:** 3

**Summary:**

This work proposes EXO-UCB that achieves sublinear regret in 2-player games while ensuring an  opponent's utility remains in a specified interval. The motivation lies in applications such as serious games where human users need to be engaged while participating in some task with an AI and may drop off if their utility is too low/high. The contributions of the work are largely theoretical; the authors model the scenario as 2-player EFG and derive an algorithm that uses an "engaging" set of strategies based on an incrementally updated confidence interval over the opponent's (fixed) strategy. The experiments on Kuhn's poker validates that the algorithm works as expected, i.e., it achieves sublinear regret while guaranteeing the utility constraints.

**Limitations And Societal Impact:**

Yes, limitations and societal impact are discussed.

**Main Review:**

Overall, the paper makes an interesting contribution towards better human-AI interaction. Given the assumptions and problem formulation, the derived results and subsequent EXO-UCB method appear sound. The generalization to infosets, i.e., the imperfect information setting, and its use to construct the engaging set was interesting to me. The experiments are sufficient to validate the overall methodology (at least up to the assumptions).

On the downside, it is difficult to judge the significance and potential impact of the work. The theoretical contributions appear divorced from the practical motivation (e.g., serious games). The assumptions in the problem formulation are unrealistic; in real settings, humans learn and do not have perfect recall. Furthermore, engaging a human is likely more complex than just introducing a bound over the obtained utilities. For example, in serious games or educational settings, a sense of accomplishment and progression is important to keep humans engaged. Potentially, the paper can be strengthened with references that support the constraint-based setup or by adding a discussion of other constraints can be formulated as bounds over obtained utility.

A second concern is about the scalability of the approach. EXO-UCB requires repeatedly solving a billinear program. The authors claim that the method is computationally low cost but a short discussion on its computational properties (e.g., how the method scales with the size of the game tree) or experimental evidence would help support the claim.

I believe this work is a good first step and I am supportive of the direction, but the paper appears preliminary at this stage. I am willing to adjust my score based on the authors' response.

### Post-Response Comments.
Thank for your response and I have raised my score to a 6. I hope the authors will modify the paper as discussed in the response. A key concern is the disconnect between the motivation and the presented results, which is shared by a majority of the reviewers; I hope that the authors will amend their motivation accordingly.

**Time Spent Reviewing:**

3

---

> ### Author Response · Authors · 2021-08-10
> **Authors’ answer to reviewer 1aGH.**
>
> We thank the Reviewer for the comments.
>
> Re: *“in real settings, humans learn and do not have perfect recall.”*:
> We believe that the case of non-learning human opponents is a natural starting point for the study of the problem of human players engagement, which is indeed a novel problem yet unexplored in the literature. Indeed, even in the case of non-learning human opponents, there are considerable challenges that need to be faced with novel and non-trivial techniques. Moreover, our results can be easily extended to the case of slowly-learning opponents by using a sliding window approach. We refer the Reviewer to the answer to Reviewer gLaM for a more extensive discussion on this. As for the perfect recall assumption, we notice that it is a technical assumption of the game and it is not needed that the human player respects this assumption, while the algorithm relies on it.
>
> Re: *“Furthermore, engaging a human is likely more complex than just introducing a bound over the obtained utilities.”*:
> We agree with the Reviewer that human engagement is not easily framed within some mathematical framework. We believe that our approach of using a bound on expected utility is well motivated, since (1) it is perhaps the easiest mathematical definition of engagement one may come up with, and (2) it is also supported by several works that take a psychological perspective on the problem (see for instance Believability Through Psychosocial Behaviour: Creating Bots That Are More Engaging and Entertaining. Believable Bots 2012: 29-68 by Bailey et al.). By the way, one can imagine a dynamic version of our constraints in which, at every round, the utility thresholds change to capture the progression mentioned by the Reviewer (e.g., increasing round by round). Interestingly, our results in terms of regret holds even in this case as they do not require that the thresholds are the same at every round (the same happens, e.g., in linear-UCB in which the space of arms can be different round by round). We also notice that our engagement constraints can be combined with other, different constraints to provide a richer language to capture engagement. We will explore that in future work and mention it in the paper.
>
> Re: *“A second concern is about the scalability of the approach. EXO-UCB requires repeatedly solving a bilinear program. The authors claim that the method is computationally low cost but a short discussion on its computational properties (e.g., how the method scales with the size of the game tree) or experimental evidence would help support the claim.”*:
> We agree with the Reviewer that this point should be clarified better. The correct statement is: “solving bilinear problems with Kuhn and Leduc poker games requires a computation time that is negligible w.r.t the time needed to get a single sample in practice for many games”. According to our experimental activity, we observed the following.
> - TS approaches based on sampling, so as to avoid the bilinarity and therefore NP-hard oracles, converge very slowly (see our results in the Appendix where we present EXO-TS) --- notice: EXO-TS is a direct adaptation of linear-TS to our problem, however, a better version of linear-TS designed ad-hoc for our problem (and developed after the submission of our paper) converges much faster than EXO-TS, but more slowly than our EXO-UCB --- so, if in practice getting a single sample requires more time than solving a bilinear problem, adopting EXO-UCB makes sense. We agree that with larger games, exact bilinear oracles are infeasible. By the way, we can use tricks to deal with them as discussed in the next point.
> - Instead of solving exactly the bilinear problem, we use in our experiments a time-limit that allows, in practice, the algorithm to return a good solution, so as to have an acceptable trade-off between computational time and solution quality. Notice that, as the number of rounds increases, the size of the confidence region reduces and, therefore, we can get a high solution quality even using a short time-limit as any feasible solution is close to the optimal one (in the degenerate case in which the confidence region collapses to a single point, the problem is not bilinear anymore). We agree that a detailed evaluation of the time-limit should be performed as the size of the games increases, but we also believe that this issue is beyond the scope of our paper. By the way, we will mention it in the experimental discussion.
>
> We agree with the reviewer that the considerations above should be clarified better in the paper, and we will do so.

---

### Official Review · Reviewer_zQhR · 2021-07-15

**Rating:** 7
**Confidence:** 3

**Summary:**


This paper addresses the problem of learning a strategy in an extensive-form game that maximizes utility gained, subject to constraints on that strategy.  These constraints are that the expected utility of the fixed opponent strategy be within some bounds with high probability on each iteration of the game. The authors proceed by precisely characterizing the region of the opponent's strategy space where the true opponent strategy lies, with high probability, given the observations of the opponents action choices thus far in the interaction.  This region is then used to generate a set of constraints on the agent's strategy, and finally, the agent's strategy is chosen to maximize its own utility, subject to the constraints and with under the optimistic assumption that the opponent will utilize also play to maximize the agent's utility (within the characterized strategy region). This strategy choice by the agent is shown to have sub-linear regret. Experimental results are provided for the domain of kuhn poker, showing that the proven bounds on opponent utility and regret of the agent utility hold in practice.

**Ethical Concerns:**

No.

**Limitations And Societal Impact:**

A negative societal impact that I can imagine is that agents could get too good at engagement and humans might not ever want to stop playing against them, but this is not necessarily a new problem with this work, I think it applies to all kinds of computer games and interactions in general.  The possibility of positive engagement benefits in serious games, for example, are well worth the effort.  The authors don't address this and I would encourage them to think about mentioning it.


**Main Review:**

This paper addresses a fairly novel problem in an effective way. It is generally well-written and clear.  I found the setting well-motivated and the potential for impact of this problem formulation and approach good.  I think that the idea of engagement with players is one that has not seen enough focus thus far within the AI/ML communities, and this work does a good job of addressing it. The work appears to be high quality, and I enjoyed reading it and thinking about the problem they are addressing.

Comments:
- I had a few minor concerns with the ways some previous work was missing that might be relevant to the paper at hand.  For example, while it is true that most efforts in extensive-form games (I am most familiar with poker) have focused on equilibrium strategies that ignore the opponent, there was a portion of the Annual Computer Poker Competition (I believe it was called the bankroll competition) where agents were evaluated based on how much they could exploit opponents, as opposed to just beating them.  I am not certain how many competitors specifically utilized other (i.e. non-equilibrium) strategies for this competition, but I know some work focused on opponent-modeling (for example N. Bard, et. al, Decision-theoretic Clustering of Strategies, AAMAS, 2015, and other work by Nolan Bard), and seems relevant for inclusion in the discussion in this paper. On a similar note, I would argue that the first major break-through in poker (for the first paragraph of the paper) was (M. Bowling, N. Burch, M. Johanson, O. Tammelin, Heads-up limit hold’em poker is solved, Science, 2015).
- Another work I am familiar with that also seems related is (Davis, T., Waugh, K., & Bowling, M. Solving Large Extensive-Form Games with Strategy Constraints. AAAI, 2019).  In this they present a method for learning an optimal strategy for extensive form games subject to convex constraints.  This seems to be the same problem faced during the final step (strategy selection) in this paper, and its relevance should be addressed.
- The experimental results that were included in the paper are solid, but a bit brief.  I would have appreciated just a bit more exposition on how the opponent strategies were generated.  In the discussion of the results, it states that Figure 1a shows the expected utility convering to 0.3, 0.28, and 0.24, but I don't see how that is the case from the figure.  The blue and green lines appear to be converging to ~0.25 and ~0.2, while the orange and yellow lines are around ~-0.2  The results do clearly show how the utility stays within the specified bounds though.
- Without thinking about it too much, I might expect a random strategy (from the constrained X) to perform more in the middle of the utility bounds, while these results show that it is closer to the worst case for the agent, within the bounds.  I wonder why this occurs?

Minor/Typos:
- In Theorem 1 there is a reference to Theorem 3.  I think that this should be Lemma 3 instead.



**Time Spent Reviewing:**

3

---

> ### Author Response · Authors · 2021-08-10
> **Authors’ answer to reviewer zQhR.**
>
> We thank the reviewer for the comments and for the interest shown in our work.
>
> Re: *“I had a few minor concerns with the ways some previous work was missing that might be relevant to the paper at hand...”*:
> Thanks for the suggestions. We will add a discussion on these related works in the final version of the paper.
>
> Re: *“Another work I am familiar with that also seems related is (Davis, T., Waugh, K., & Bowling, M. Solving Large Extensive-Form Games with Strategy Constraints. AAAI, 2019). In this they present a method for learning an optimal strategy for extensive form games subject to convex constraints. This seems to be the same problem faced during the final step (strategy selection) in this paper, and its relevance should be addressed.”*:
> The work is indeed related to ours and we will discuss their differences. The main one is the fact that in the strategy selection part of our algorithm we have a max-max problem (over X^t and Y^t), while this work considers a constrained max-min problem.
>
> Re: *“The blue and green lines appear to be converging to ~0.25 and ~0.2, while the orange and yellow lines are around ~-0.2 The results do clearly show how the utility stays within the specified bounds though.”*:
> We will surely expand the description of the results in order to make them clearer. More specifically we will add that that point convergence to the optimal values requires more iterations.
>
> Re: *“Without thinking about it too much, I might expect a random strategy (from the constrained X) to perform more in the middle of the utility bounds, while these results show that it is closer to the worst case for the agent, within the bounds. I wonder why this occurs?”*:
> Our conjecture on why this happens is that the random strategy is indeed constrained, while the random strategy of the human is a sort of low-entropy random strategy. We expect a value close to zero only in the case of unconstrained random vs random strategy.

---

### Official Review · Reviewer_gLaM · 2021-07-22

**Rating:** 5
**Confidence:** 4

**Summary:**

The paper addresses the problem of maintaining the engagement of a human player when interacting with an AI agent within the context of a game. To do so, the paper proposes a method by which an AI agent learns online the behaviour of the human agent and acts in such a way to keep the reward of the human within some interval so as to ensure the game is neither too difficult nor too easy. Theoretical results are established that bound the probability that the reward lies within this interval as well as regret bounds for the performance of the AI agent.


**Limitations And Societal Impact:**

I think this is covered satisfactorily.

**Main Review:**

I think the paper is well motivated and the problem undoubtedly has important applications. The authors have established nice results that bound the regret and violations. Overall, while I find the framework to be skillfully developed with interesting results, it feels to me to be poorly matched with the problem. Perhaps the authors could consider if there are applications within safety problems e.g. humans in the loop with assistive AI agents which require this sort of treatment (but without the zero-sum assumption).

**Detailed Comments**

In the authors’ treatment of the problem, the human is assumed to have a fixed stochastic behaviour and not to learn over time. Given this, I find the setup somewhat incompatible with the aims of the paper and the problem. In particular, part of what I would think keeps a human engaged is progressive learning at the correct pace – including a second learner (with some means of modelling human behaviour e.g. a boundedly rational learner) seems like a requirement for this setting.

The paper also does not include experiments to test the main claim that the framework promotes prolonged engagement. This further makes it difficult to confidently understand if the modelling assumptions of the paper are suitable for the task at hand. While I understand performing experiments that involve human agents carries its difficulties, considering that the entire premise of the paper is that there is a need for algorithms that keeps humans engaged, it seems to me that human experiments should have been included.

Additionally, though the problem being targeted is one in which two interested players engage in a strategic interaction, the human is fixed the strategic interaction aspect is removed. This somewhat renders the exposition on extensive form games, which takes up an appreciable part of the paper somewhat redundant .

The theoretical analysis contains nice results, many of the key results are stated without accompanying explanation of their importance and relevance which undermines the analysis somewhat.

**Questions**

*Why can’t we just insert a term in the agent’s payoff which penalises it for taking actions that give the other agent a reward outside of a given range?

*If the goal of the agent is to simply ensure that the human’s payoff lies within a certain range – why do we need to care about the zero-sum objective (i.e. why can’t we just maximise engagement)?

*How does using a bandit method compare to tree search methods? Particularly in the way of scaling with the size of the tree.


**Time Spent Reviewing:**

7

---

> ### Author Response · Authors · 2021-08-10
> **Authors’ answer to reviewer gLaM.**
>
> We thank the Reviewer for the comments. Summarily, we would like to invite the Reviewer to reconsider our paper as our opinion is that he/she did not perfectly capture its spirit. More precisely, we aim at designing an algorithm with ***theoretical guarantees*** capable of learning in online fashion to play against a human under constraints on her/his utility loss. This problem is novel, and non-trivial, even when the human has a fixed stochastic strategy, and any analysis when (even) the human player is learning cannot be carried out without first studying the case with fixed strategy. Furthermore, our techniques can be easily extended  to non-adversarial settings in which the human changes smoothly her/his strategy (e.g., using a sliding-window approach). Notice that some tricks suggested by the Reviewer may not lead to theoretical guarantees (e.g., introducing penalties in the payoffs) or can allow us to deal with large problems (MC sampling/tree search), but they do not deal with utility-loss constraints and/or regret minimization guarantees. We agree that the comments raised by the Reviewer should be mentioned and clarified in the paper, but they do not affect the novelty and significance of our contributions.
>
> Re: *“Overall, while I find the framework to be skillfully developed with interesting results, it feels to me to be poorly matched with the problem. Perhaps the authors could consider if there are applications within safety problems e.g. humans in the loop with assistive AI agents which require this sort of treatment (but without the zero-sum assumption).”*:
> We thank the Reviewer for suggesting us possible applications of our framework; such settings seem to fit nicely in our framework and we will surely consider them for future works. However, we would like to stress that the problem we face in our paper (that is, keeping human players engaged) is very interesting from both a theoretical and a practical perspective, and yet it did not receive any attention from the AI community, as also observed by Reviewer zQhR. Moreover, we would like to point to the attention of the Reviewer that the zero-sum assumption is indeed not technically needed in our framework, but it was added for simplifying exposition as the experimental evaluation only involves classical zero-sum game testbeds. In the final version of the paper, to avoid any confusion, we will drop such an assumption and state the results in the general-sum setting.
>
> Re: *“In the authors’ treatment of the problem, the human is assumed to have a fixed stochastic behaviour and not to learn over time. Given this, I find the setup somewhat incompatible with the aims of the paper and the problem. In particular, part of what I would think keeps a human engaged is progressive learning at the correct pace – including a second learner (with some means of modelling human behaviour e.g. a boundedly rational learner) seems like a requirement for this setting.”*:
> Notice that the problem that we face in our paper (where the opponent exhibits a fixed stochastic behavior) has never been studied in the literature and we believe it constitutes a natural starting point for studying the problem of engaging human players. Moreover, even our setting with non-learning opponents poses considerable difficulty challenges, which require some novel and non-trivial techniques in order to be solved. Nevertheless, we agree with the Reviewer that more complex models of engagement are really interesting for future works (though out of the scope of our paper). For instance, if one would need to model the case of slowly learning opponents, a natural extension of our framework is to adopt a sliding window approach, where each window corresponds to some kind of stage of the human learning process. We will add a discussion on this and other possible extensions in the final version of the paper.
>
> Re: *“The paper also does not include experiments to test the main claim that the framework promotes prolonged engagement.”*:
> Let us remark that capturing engagement by forcing utility constraints is customary in the literature: see, for instance, (Believability Through Psychosocial Behaviour: Creating Bots That Are More Engaging and Entertaining. Believable Bots 2012: 29-68 by Bailey et al.). By the way, we are currently working on more extensive experiments that will also involve some tests involving human players. Since these experiments usually require considerable time, we plan to have them available for possible future work along this line of research. The aim of this paper is mainly to introduce the problem (novel in itself) and to describe the mathematical framework put in place to solve, so we believe that experiments on standard testbeds are indeed sufficient to validate our results.
>
> Re: *“Additionally, though the problem being targeted is one in which two interested players engage in a strategic interaction, the human is fixed the strategic interaction aspect is removed. This somewhat renders the exposition on extensive form games, which takes up an appreciable part of the paper somewhat redundant”*:
> Notice that, even if the opponent has a fixed behaviour, the structure of extensive-form games is needed for our results. In particular, our strategy estimation step deeply relies on the particular structure of the sequence form of extensive-form games.
>
> Re: *“Why can’t we just insert a term in the agent’s payoff which penalises it for taking actions that give the other agent a reward outside of a given range?”*:
> Modifying the payoff of the game to have a high probability of respecting the engagement constraints is non-trivial, as you need such a modification to change over time to have the constraints respected at each iteration. It is not clear how this can be carried out in our setting.
>
> Re: *“If the goal of the agent is to simply ensure that the human’s payoff lies within a certain range – why do we need to care about the zero-sum objective (i.e. why can’t we just maximise engagement)?”*:
> The goal is the maximization of a given utility function while guaranteeing that the human is sufficiently engaged (and this is captured by controlling the loss of the human to be in the range). The optimization problem is the same both in the zero-sum and general-sum games. We focused on zero-sum just for the sake of presentation, and because a natural example is that of repeated poker. We agree that this point should be clarified better in the paper, and we will do so.
>
> Re: *“How does using a bandit method compare to tree search methods? Particularly in the way of scaling with the size of the tree.”*:
> Understanding the differences between using a single bandit method and using UCB-Tree (or MCTS) is an interesting research question, which is not explored in the literature for our specific setting. We are currently studying it from a theoretical point of view for another work.

---

### Decision · Program_Chairs · 2021-09-27

**Decision:**

Accept (Poster)

**Comment:**

The paper addresses the problem of human engagement in repeated games, developing a novel algorithm that guarantees payoffs to the human can be kept within particular bounds over time with high probability. All reviewers agree that the problem is interesting and relevant. The topic is particularly important as the number of applications of human-computer interaction rise.

After the response there was some discussion. There are a few problems with the paper in its current form: (a) the problem is not well-motivated, (b) the contribution is mainly theoretical and not practically tied to human engagement since the "human strategy" is fixed, (c) the relationship between human engagement and utilities being within a certain range is not firmly backed by real evidence, and (d) the experimental results are not well-matched to the main problem being motivated by the paper (i.e. no evidence of actual increased human engagement).

The reviewers appreciated the thorough responses and satisfied by clarifying of technical points. Three of the four reviewers agreed post-response that (a) is easily fixed in a final copy. This is also true for (c) by adding (and perhaps discussing) the references mentioned in the responses more prominently. (b) and (d) are valid outstanding criticisms, which make the impact unclear at this point. But, a proper theoretical investigation on a fixed policy does act as an important stepping stone toward these eventual goals. Taking this into account, combined with the novelty and interest this paper could generate, the positives outweigh the shortcomings. Still, it is important to take all the critical feedback strongly into consideration when revising the paper.